# Targeting CD123 in blastic plasmacytoid dendritic cell neoplasm using allogeneic anti-CD123 CAR T cells

Tianyu Cai [1], Agnès Gouble[2], Kathryn L. Black[3], Anna Skwarska[1], Ammar S. Naqvi [3], Deanne Taylor [4], Ming Zhao[5], Qi Yuan [6], Mayumi Sugita [7], Qi Zhang [1], Roman Galetto[2], Stéphanie Filipe[2], Antonio Cavazos[1], Lina Han[1], Vinitha Kuruvilla[1], Helen Ma[1], Connie Weng[1], Chang-Gong Liu[8], Xiuping Liu[8], Sergej Konoplev[9], Jun Gu [5], Guilin Tang[9], Xiaoping Su[6], Gheath Al-Atrash[10], Stefan Ciurea[10], Sattva S. Neelapu [11], Andrew A. Lane [12], Hagop Kantarjian [1], Monica L. Guzman [7], Naveen Pemmaraju[1], Julianne Smith[2], Andrei Thomas-Tikhonenko[3] & Marina Konopleva [1]✉

Blastic plasmacytoid dendritic cell neoplasm (BPDCN) is a rare hematologic malignancy with poor outcomes with conventional therapy. Nearly 100% of BPDCNs overexpress interleukin 3 receptor subunit alpha (CD123). Given that CD123 is differentially expressed on the surface of BPDCN cells, it has emerged as an attractive therapeutic target. UCART123 is an investigational product consisting of allogeneic T cells expressing an anti-CD123 chimeric antigen receptor (CAR), edited with TALEN® nucleases. In this study, we examine the antitumor activity of UCART123 in preclinical models of BPDCN. We report that UCART123 have selective antitumor activity against CD123-positive primary BPDCN samples (while sparing normal hematopoietic progenitor cells) in the in vitro cytotoxicity and T cell degranulation assays; supported by the increased secretion of IFNγ by UCART123 cells when cultured in the presence of BPDCN cells. UCART123 eradicate BPDCN and result in long-term disease-free survival in a subset of primary patient-derived BPDCN xenograft mouse models. One potential challenge of CD123 targeting therapies is the loss of CD123 antigen through diverse genetic mechanisms, an event observed in one of three BPDCN PDX studied. In summary, these results provide a preclinical proof-of-principle that allogeneic UCART123 cells have potent anti-BPDCN activity.

[1] Department of Leukemia, The University of Texas MD Anderson Cancer Center, Houston, USA. [2] Cellectis SA, Paris, France. [3] Department of Pathology and Laboratory Medicine, Children's Hospital of Philadelphia and Perelman School of Medicine at the University of Pennsylvania, Philadelphia, PA, USA. [4] Department of Biomedical & Health Informatics, Children's Hospital of Philadelphia and Perelman School of Medicine at the University of Pennsylvania, Philadelphia, PA, USA. [5] School of Health Professions, The University of Texas MD Anderson Cancer Center, Houston, TX, USA. [6] Department of Bioinformatics and Computational Biology, The University of Texas MD Anderson Cancer Center, Houston, TX, USA. [7] Department of Medicine, Division of Hematology & Medical Oncology, Weill Cornell Medicine, New York, NY, USA. [8] Department of Experimental Therapeutics, The University of MD Anderson Cancer Center, Houston, TX, USA. [9] Department of Hematopathology, The University of MD Anderson Cancer Center, Houston, TX, USA. [10] Department of Stem Cell Transplantation and Cellular Therapy, The University of MD Anderson Cancer Center, Houston, TX, USA. [11] Department of Lymphoma and Myeloma, The University of MD Anderson Cancer Center, Houston, TX, USA. [12] Dana-Farber Cancer Institute, Boston, MA, USA. ✉email: mkonople@mdanderson.org

Blastic plasmacytoid dendritic cell neoplasm (BPDCN) is a rare, aggressive hematologic malignancy that originates from the precursors of plasmacytoid dendritic cells[1,2]. Presenting signs of BPDCN include skin nodules and tumors, lymph node and splenic enlargement, central nervous system involvement, and circulating leukemia and/or bone marrow infiltration[3]. The biology of BPDCN is not completely understood; patient outcomes have historically been poor and no standard of care has been established[4]. Median survival has been reported to be approximately one to one and a half years, or even shorter in patients who have disseminated disease[5,6]. Novel therapies are therefore urgently needed for patients with BPDCN.

CD123, the interleukin 3 receptor alpha chain (IL3RA), is the primary low-affinity subunit of the interleukin 3 receptor. CD123 is normally expressed on some endothelial cells, monocytes, plasmacytoid dendritic cells, basophils, and myeloid progenitors. However, one of the major early breakthroughs in BPDCN was the discovery that CD123 is overexpressed in essentially all cases[7–9]. CD123 has emerged as an attractive therapeutic target given its differential expression on BPDCN cells[10–13], as it is expressed at markedly higher levels on BPDCN blasts than on cells in the normal hematopoietic stem cell compartment[14,15]. The first CD123-targeted therapy tagraxofusp (SL-401, a diphtheria toxin interleukin-3 fusion protein) was recently approved by the FDA specifically for treatment of this rare disease[16]. Last year, an antibody-drug conjugate IMGN632 has received from FDA a fast-track designation for BPDCN therapy, further highlighting the feasibility, importance, acceptable safety, and clinical impact of targeting CD123.

UCART123 is a genetically modified allogeneic T cell product manufactured from healthy donor cells. These modified T cells (CD123CAR+_TCRαβ-_T cells) express (i) a second-generation chimeric antigen receptor (CAR; CD123 scFv-4-1BB-CD3ζ) directed against CD123 and (ii) the "safety switch" RQR8 depletion ligand, combining epitopes from both CD34 and CD20 antigens, which confers susceptibility to rituximab[17,18] (Supplementary Fig. 1). CD123CAR and RQR8 are encoded by the same lentiviral vector (Fig. 1a). Expression of the T cell receptor (TCR) αβ on these cells is abolished through inactivation of the TCRα constant (TRAC) gene using TALEN® gene editing technology. We have previously reported that the TRAC KO has no impact on CAR T-cell activity however was able to eliminate the potential for graft versus host reactions[17].

In this study, we demonstrate that allogeneic UCART123 therapy resulted in eradication of BPDCN in vitro and in long-term disease-free survival in a subset of primary BPDCN PDX experiments.

## Results

### UCART123 cells result in specific killing of CAL-1 BPDCN cells.
To test the in vitro activity of UCART123 against BPDCN cells, we first used BPDCN cell line CAL-1. The antitumor activity of UCART123 on these tumor cells was assessed in vitro using the cytotoxicity, degranulation, and IFNγ release assays.

The cytotoxic activity of UCART123 was evaluated by co-culturing UCART123 cells with CAL-1 cells at different effector to target (E:T) ratios, followed by flow cytometry analysis to identify viable target cells. Percentage of cell viability was compared to that after co-culture with TCRαβ KO T cells (non-transduced (CAR-), TCRαβ- T-cells), and the percentage of specific cell lysis was calculated. In this assay, UCART123 cells targeted CAL-1 cells in a dose-dependent fashion, reaching high levels of specific lysis ($80.3 \pm 5.5\%$) at 10:1 E:T ratio (Fig. 1b and Supplementary Fig. 2a).

The cytotoxic capacity and activation of UCART123 against CAL-1 cells was confirmed by the expression of CD107α as a marker of T cell degranulation (Fig. 1c). CD107α was expressed by UCART123 cells in the presence of BPDCN cells in the same range as observed with nonspecific stimulation (PMA/Ionomycin).

The secretion of cytokines, specifically IFNγ, by UCART123 cells in response to specific stimulation by BPDCN cells was quantified. The results show that UCART123 cells, when co-cultured with CAL-1 cells, secrete high levels of IFNγ, IL-2, IL-5, IL-13, and TNFα (Fig. 1d and Supplementary Fig. 2b–f).

To analyze relationship between UCART123 efficacy and CD123 cell surface expression, we used a series of (non-BPDCN) cell lines with varying levels of CD123, with Daudi cell line not expressing the target. No cytotoxic activity of UCART123 was elicited against Daudi cells, while there was a clear trend towards increasing degranulation, cytotoxic activity and IFNγ release by CART cells against AML cell lines with low, medium and high CD123 levels (Supplementary Fig. 3).

### UCART123 cells result in specific killing of BPDCN primary samples.
To verify the clinical relevance of CD123 as a target for immunotherapy in BPDCN, we first analyzed the bone marrow (BM) of 8 patients with newly diagnosed BPDCN to determine CD123 expression in comparison to BM blasts of 28 patients with newly diagnosed acute myeloid leukemia (AML) and 13 AML patients in remission. CD123 expression levels were significantly higher in BPDCN (gated on CD45 + cells, mean fluorescence intensity [MFI] range 3,484–17,937) than in remission marrows (gated on CD34 + cells, MFI range 232–845; $p \leq 0.01$) and in AML blasts (gated on CD45 + cells, MFI range 360–5073; $p \leq 0.01$) (Supplementary Fig. 4). Its ubiquitous and high expression in BPDCN samples suggests that CD123 represents a rational target for therapeutic development in BPDCN.

Additional BPDCN samples freshly procured from patients or viably stored at MDACC Leukemia sample bank were obtained for in vitro analysis (Table 1). All samples expressed CD123 although at various levels (MFI range, 1863–136,399) (Fig. 2a). Cytotoxicity assay results showed UCART123 activity against BPDCN samples with high level of specific cytotoxicity against 5 of the 8 samples evaluated (Fig. 2b and Supplementary Fig. 5). In a degranulation assay, CD107α was detected at the surface of UCART123 cells in the presence of BPDCN samples, in the same range as observed with nonspecific stimulation (PMA/ionomycin, Fig. 2c). UCART123 cells, when co-cultured with BPDCN cells, secreted significantly higher levels of IFNγ than when co-cultured with negative control Jurkat cells (Fig. 2d). Furthermore, UCART123 was shown to have minimal toxicity against normal hematopoietic cells in vitro in both a cytotoxicity and a colony formation assay (Supplementary Fig. 6).

### In vivo activity of UCART123 cells against BPDCN in PDX experiments.
To evaluate the antitumor activity of UCART123 cells in vivo, we established patient-derived xenograft (PDX) models in NSG and NSG-SGM3 (NSGS) mice using cells from BPDCN patients (BPDCN-1 to -3 in Table 1). In the first PDX experiment (PDX-1, using sample BPDCN-1), mice were randomized upon tumor engraftment into 4 treatment groups: vehicle, $10 \times 10^6$ TCRαβ KO T cells, $3 \times 10^6$ or $10 \times 10^6$ UCART123 cells; each treatment administered as a single tail vein injection (Fig. 3a). All mice from the control groups (vehicle or TCRαβ KO T cells) died by day 53 post BPDCN injection, while UCART123 treatment prolonged mice survival. Furthermore, a significant fraction of the mice treated with UCART123, at both doses, remained alive at the end of the study (day 300) (Fig. 3b).

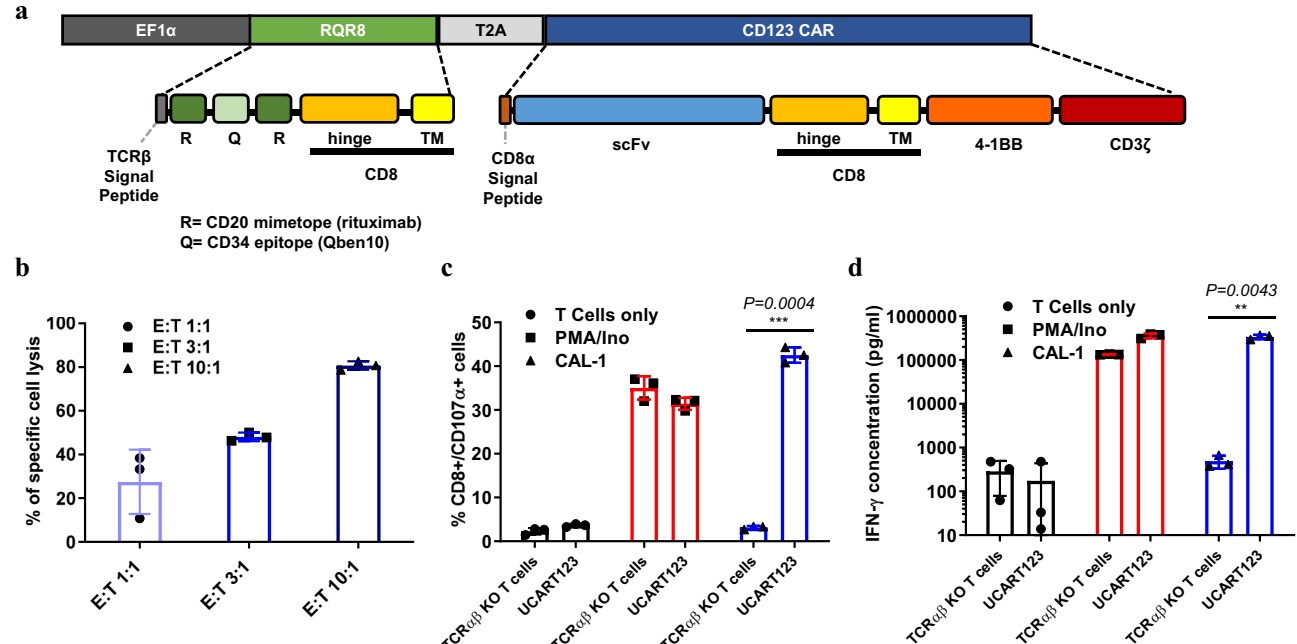

**Fig. 1 Cytotoxicity of UCART123 against CAL-1 BPDCN cells in vitro. a** UCART123 cells express (i) a second-generation chimeric antigen receptor (CAR; CD123 scFv-4-1BB-CD3z) directed against CD123 and (ii) the "safety switch" RQR8 depletion ligand. **b** CAL-1 BPDCN cells were co-cultured with either non-transduced TCRαβ-deficient (TCRαβ KO) T cells or with UCART123 cells at various effector: target (E:T) ratios indicated for 16 h. The specific cytotoxic activity of UCART123 against CAL-1 target cells was calculated. Each point represents the data obtained from triplicate experiments, and the mean ± SD value is presented. **c** CD107α degranulation was measured by flow cytometry gating on CD8+ cells after co-culture with UCART123 or TCRαβ KO T cells at E:T = 1:1 for 6 h. "T cells only" correspond to the basal activity of unstimulated T cells, while PMA/Ion corresponds to the signal observed upon nonspecific stimulation by phorbol myristate acetate and ionomycin. Data represent n = 3 biological replicates and mean ± SD of triplicates. Significance was determined using unpaired two-tailed t-test annotated as ***$P \leq 0.001$. **d** CAL-1 cells were co-cultured with either TCRαβ KO T cells or with UCART123 cells at E:T = 1:1 for 25 h. IFNγ levels were determined by the BioLegend LEGENDplex assay. Data represent n = 3 biological replicates and mean ± SD of triplicates. Significance was determined using unpaired two-tailed t-test annotated as **$P \leq 0.01$. Source data are provided as a Source data file.

Measurement of the percentage of BPDCN cell engraftment by flow cytometry indicated that both doses of UCART123 allowed control of tumor progression, and that the circulating tumor burden was reduced from day 34 after UCART123 injection (Fig. 3c). The presence of circulating tumor cells was evaluated in the long-term surviving mice (i.e., the UCART123 treatment groups) on days 267 and 279 post BPDCN cell injection (Table 2), showing efficient and durable control of tumor progression in surviving mice. In a mouse from the $10 \times 10^6$ UCART123 treatment group expiring on Day 150, no BPDCN cells (hCD123/hCD56 double positive cells) were detected in the BM or spleen (Supplementary Fig. 7), indicating that the cause of death was not likely due to BPDCN progression.

The presence of UCART123 cells was assessed in the blood, spleen, and BM of one mouse from the $10 \times 10^6$ UCART123 treatment group sacrificed on day 78 (57 days after UCART123 cell injection). UCART123 cells could be detected in the spleen (16.4% CAR+ cells among viable cells; Supplementary Fig. 8) and in the bone marrow (1.1% CAR+ cells; Supplementary Fig. 8), indicating long-term persistence.

These results indicate that UCART123 treatment led to long-term survival and elimination of BPDCN in this primary BPDCN xenograft model.

**Elevated cytokine levels correlate with tumor burden.** In a second PDX experiment (PDX-2, using sample BPDCN-3), mice were similarly randomized upon tumor engraftment into four treatment groups: vehicle, $10 \times 10^6$ TCRαβ KO T cells, $3 \times 10^6$ or $10 \times 10^6$ UCART123 cells, each treatment administered as a

single tail vein injection (Fig. 4a). All mice treated with UCART123 unexpectedly died 5–7 days after UCART123 injection (day 28 after BPDCN cell injection), while the mice in the vehicle group survived slightly longer (Fig. 4a, b). Circulating hCD123 + cells indicating high tumor burden were found in all cohorts (Fig. 4c) by day 23. A high level of IFNγ and IL-2 was detected in peripheral blood samples from the mice treated with UCART123 cells 2 days following UCART123 injection (Fig. 4d and Supplementary Fig. 9a). These results suggest that a cytokine surge from activated UCART123 and/or tumor lysis syndrome after UCART123 administration may have contributed to the premature death of mice treated with UCART123.

We therefore repeated the experiment, this time administering the treatment at an earlier time point, day 14 after BPDCN cell injection (Fig. 4e). All mice in the control groups (treated with vehicle or TCRαβ KO T cells) experienced tumor progression by day 34 after primary BPDCN cell injection and were sacrificed. In contrast, the cohorts of mice treated with UCART123 had extended survival, with those treated with the highest dose ($10 \times 10^6$) of UCART123 cells all alive at the end of the study (day 210), with no detectable circulating tumor cells (Fig. 4f, g). Mice treated with UCART123 experienced cytokine release 2 days after UCART123 injection, but the concentrations of IFNγ and IL2 in the peripheral blood were at least 10-fold lower than that measured in the preceding experiment with the same PDX-2 experiment (Fig. 4h and Supplementary Fig. 9b). This supports the hypothesis that cytokine release syndrome contributed to the premature deaths of the UCART123-treated PDX-2 mice.

To further evaluate if the RQR8 targeting could lead to CAR elimination in vivo and dampen the cytokine release syndrome,

**Table 1 Characteristics of primary BPDCN patients' samples.**

| Identifier (sample source) | Cytogenetic abnormality | Mutations | Clinical Status | Blast % |
|---|---|---|---|---|
| BPDCN-1 (PB) | ND | ASXL1, JAK2, TET2, NRAS | Relapse | 46 |
| BPDCN-2 (BM) | Add(17)(p11.2), TP53 deletion by FISH | ND | Relapse | 10 |
| BPDCN-3 (BM) | 46,XY,t(1;6)(p21;p36.3), del(5)(q13q33), der(7)t(1;7)(q12;p22), del(11)(q13q23),del(12)(p11.2p13), add(15)(q15) | None identified | Relapse/ refractory | 60 |
| BPDCN-4 (BM) | 45, XY, t(1;9)(p34;q32), del(6)(q16q27), der(6)t(3;6)(q26;p25), -21, add(21)(p13)[9]/46, XY[11] | TET2 | Relapse | 64 |
| BPDCN-5 (PB) | 46,XY, +1, add(1)(p13),der(1)dup(1)(q21q32)add(1)(q42),add(2)(p24),-4 | MPL, TET2 | Relapse | 22 |
| BPDCN-6(BM) | Add(12)(p11.2), -13, -21, +2-3mar[cp6]/46, XY | TET2, IDH2 | Newly diagnosed | 72 |
| BPDCN-7 (PB) | diploid | ND | Newly diagnosed | 32 |
| BPDCN-8 (BM) | 46,XY,del(9)(q21)[8]/46,XY[3] | TP53 x2, ZRSR2, IDH2 | Newly diagnosed | 90 |

All samples were obtained from patients of male gender.
*PB* peripheral blood, *ND* not determined, *BM* bone marrow, *FISH* fluorescence in situ hybridization.

**Table 2 Circulating tumor (BPDCN) burden in mice after treatment with UCART123.**

| | Circulating tumor burden at D267 (CD123+ CD56+) (%) | Circulating tumor burden at D279 (CD123+ CD56+) (%) |
|---|---|---|
| UCART123 3 × 10^6 group | 0.1 | 0.0 |
| | 0.0 | 0.0 |
| | 0.2 | 0.0 |
| UCART123 10 × 10^6 group | 0.0 | 0.1 |
| | 0.1 | 0.0 |
| | 0.2 | 0.0 |
| | 0.1 | 0.0 |
| | 0.0 | 0.0 |
| | 0.2 | 0.0 |
| | 0.1 | 0.0 |

Circulating tumor burden was measured by flow cytometry in peripheral blood samples from mice collected on the indicated days (post tumor cell injection), using 9F5 monoclonal CD123-PE antibody (555644, BD Pharmingen) and B159 monoclonal CD56-APC antibody (555518, BD Pharmingen) gating on viable cells (DAPI(−)).

we administered rituximab 10 mg/kg i.p. 2 days after UCART injections (day 19) during 5 days (Supplementary Fig. 10a). Rituximab administration post CAR-T therapy decreased the amount of CAR-T cells in the murine spleens and bone marrow (Supplementary Fig. 10b), supporting the functionality of the safety switch in vivo. IFNγ levels were similarly reduced in peripheral blood from the mice treated with Rituximab post CAR-T therapy (Supplementary Fig. 10c). These results confirm that RQR8 targeting by rituximab leads to CAR-T elimination in vivo associated with abrogation of cytokine production.

**Loss of CD123 leading to escape from UCART123 therapy and early relapse in one of the PDX experiments**. In the PDX-3 experiment (sample BPDCN-2), which received treatments similar to those administered to PDX-1 and PDX-2 experiments, all vehicle-treated mice succumbed to BPDCN by day 49 after primary BPDCN cell injection, because of high tumor burden. hCD45+/CD123+CD56+ BPDCN cells were detected in the spleen and BM. Mice treated with UCART123 displayed significantly extended survival, as in the PDX-1 and PDX-2 experiments, to days 104–241 (Fig. 5a, b). IFNγ was detected in peripheral blood samples from the mice treated with UCART123 cells 2 days following T cell injection (Supplementary Fig. 11). However, all UCART123-treated mice experienced BPDCN relapse at days 90–155. Flow cytometry analysis showed that the relapses (in all 3 mice tested/cohort) from 3 × 10^6 UCART123 and 10 × 10^6 UCART123 cohorts were associated with emergence of CD123-negative circulating BPDCN clones (95–96% CD123 negative) (Fig. 5c). To determine if UCART123 were present at the time of relapse, we performed digital droplet PCR (ddPCR). Spleen cells were collected from vehicle or UCART123-treated mice (1 × 10^6 UCART123 and 10×10^6 UCART123) at death. UCART123 cells were readily detectable in several of the mice that died from CD123 negative relapsed tumor (Sample 13, Sample 16 from 1 × 10^6 UCART123 group, and sample 5 from 10 × 10^6 UCART123 group, Supplementary Table 1), consistent with relapse due to the loss of the target and not because of lack of persistence of UCART123 cells.

Since relapses at a low UCART123 dose level were associated with CD123 + disease recurrence, we injected secondary recipient mice with 2 × 10^6 relapsed BPDCN tumor cells (CD123+ tumor cells collected from the spleen of a mouse from the group treated with 1 × 10^6 UCART123 in the initial PDX-3 experiment), and randomized upon engraftment into vehicle or 1 × 10^6 UCART123 treatment cohorts (Supplementary Fig. 12a). On day 38 after

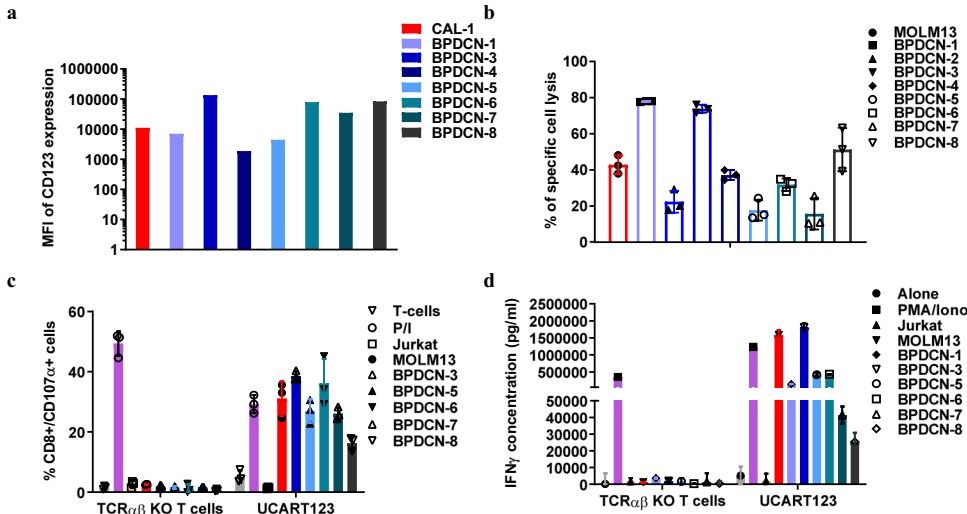

**Fig. 2 Antitumor activity of UCART123 against primary BPDCN samples in vitro. a** Expression of CD123 in CAL-1 cells and BPDCN patient samples was evaluated for 7 out of 8 samples by flow cytometry using 7G3 monoclonal anti-CD123-FITC antibody after gating on viable cells (DAPI−). **b** Specific cytotoxic activity against target cells (CD123+ MOLM13 cells and primary BPDCN samples) upon co-culture for 16 h with either UCART123 cells or TCRαβ KO T cells at E:T = 10:1. Each point represents the data obtained from triplicate experiments, and the mean ± SD value is presented. **c** CD107α degranulation was measured by flow cytometry after gating on CD8+ cells. "T cells" correspond to the basal activity of unstimulated T cells, and PMA/Ion corresponds to the signal observed upon nonspecific PMA/ionomycin stimulation. Degranulation activity upon co-culture of UCART123 or TCRαβ KO T cells with CD123- Jurkat cells as well as CD123+ MOLM13 cells and five different primary BPDCN samples is shown. Each point represents the data obtained from triplicate experiments, and the mean ± SD value is presented. **d** IFNγ release upon co-culture of UCART123 cells or TCRαβ KO T cells with CD123− or CD123+ cells at E:T = 1:1 for 25 h. IFNγ levels were determined using the BioLegend LEGENDplex assay. PMA/ionomycin was used as a positive control. Each point represents the data obtained from triplicate experiments, and the mean ± SD value is presented. BPDCN-1, 2, 4 were not tested in all assays due to limited cell numbers. Source data are provided as a Source data file.

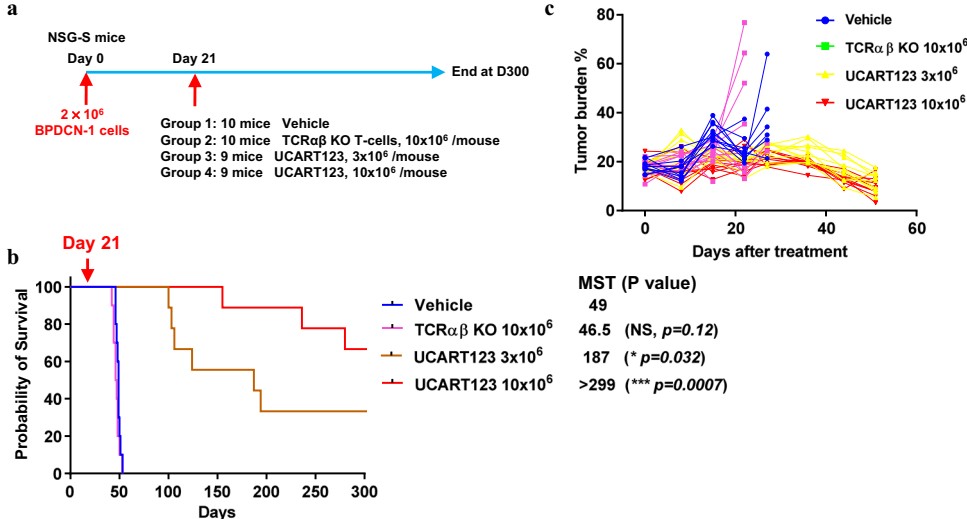

**Fig. 3 UCART123 treatment results in long-term survival of primary BPDCN PDX. a** Experimental design. NSGS mice were injected intravenously with 2 × 10⁶ cells from BPDCN-1 (68.4% CD123+) at day 0. Engraftment was confirmed by analysis of peripheral blood on Day 21 post tumor cell injection, mice were divided randomly into 4 treatment groups (*n* = 9–10 mice/group) and received treatment as follows: vehicle; 10 × 10⁶ TCRαβ KO T cells; 3 × 10⁶ UCART123 cells; or 10 × 10⁶ UCART123 cells via single tail vein injection. **b** Survival of the mice in indicated treatment groups was estimated by the Kaplan-Meier method. Treatment with UCART123 resulted in significant survival extension. MST = median survival time (in days). Significance was determined using unpaired two-tailed t-test annotated as "NS" (not significant, $p ≥ 0.05$), *$P ≤ 0.05$, ***$P ≤ 0.001$. **c** Circulating tumor burden was measured by flow cytometry in peripheral mouse blood samples collected on the indicated days (post treatment). Presence of human tumor cells was analyzed using cell surface expression of hCD123 among (DAPI-) viable cells. Source data are provided as a Source data file.

BPDCN cell injection, all the mice in the vehicle group were sacrificed because of tumor progression. The mice in the UCART123 treatment group died on days 78–99; thus, UCART123 treatment extended survival in mice engrafted with relapsed BPDCN cells (Supplementary Fig. 12b). However, similar to the study with original PDX-3 in cohorts receiving high doses of UCART123, tumor progression and death in all mice of the UCART123 low dose treatment group was now associated with emergence of CD123 negative BPDCN clones (Supplementary Fig. 12c).

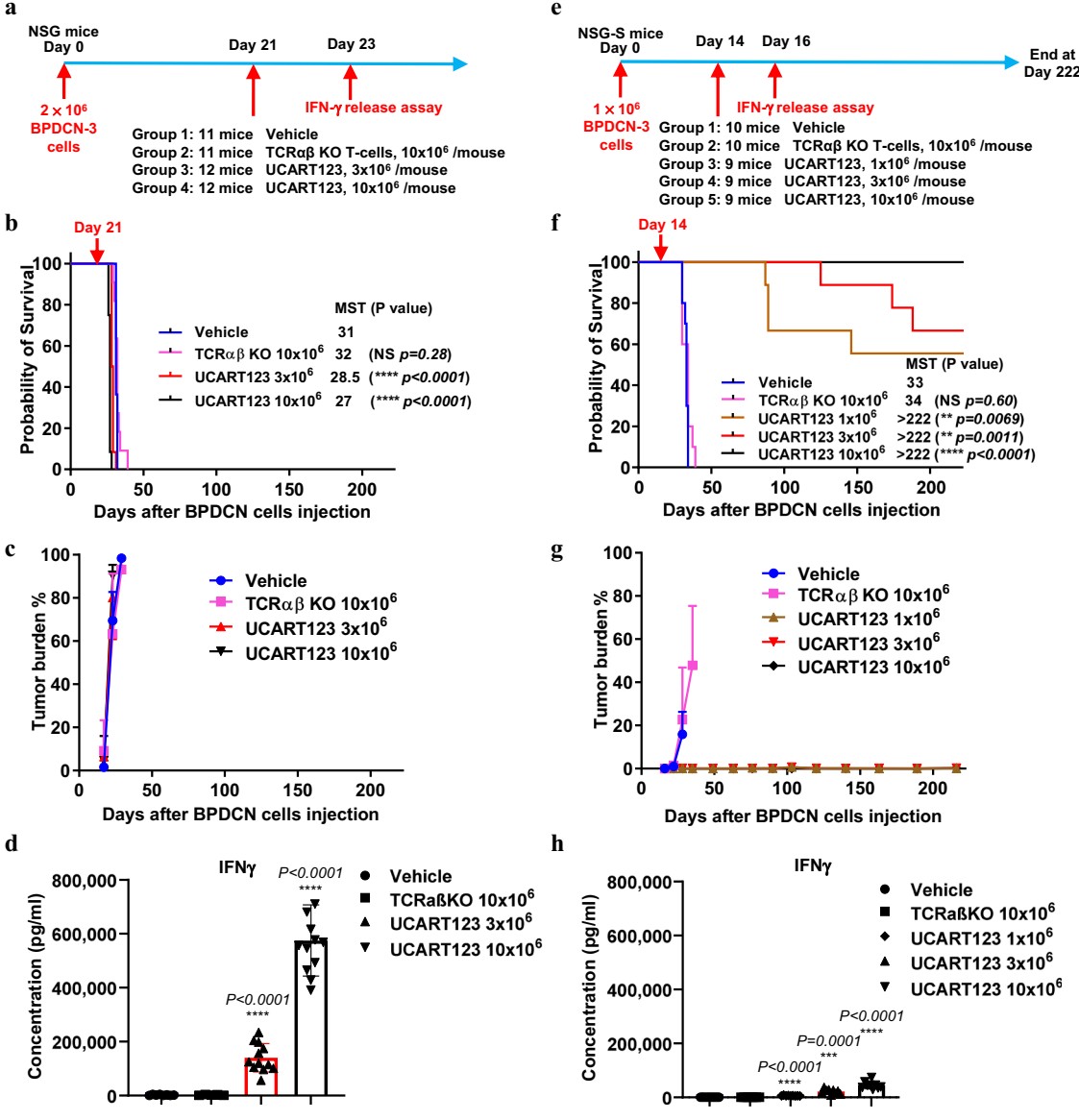

**Fig. 4 Evidence of severe cytokine release at high tumor burden in the PDX-2 BPDCN experiment. a** Experimental design using BPDCN-3 cells. When engraftment was confirmed on day 21 after tumor cell injection (0.1–23.5% circulating BPDCN), mice were randomized into 4 treatment groups (n = 11-12 mice/group) and received treatment as follows: vehicle; $10 \times 10^6$ TCRαβ KO T cells; $3 \times 10^6$ UCART123 cells; or $10 \times 10^6$ UCART123 cells by single tail vein injection. **b** Survival among the treatment groups was estimated by the Kaplan-Meier method. MST = median survival time (in days). **c** High baseline and rapidly progressing tumor burden in all cohorts. The mean ± SD value is presented. **d** Mice in the UCART123-treated groups, which had a high tumor burden when treatment began, died earlier than control mice, showing high levels of IFNγ in peripheral blood samples at Day 2 following T cells injection. The mean ± SD value is presented. **e** Experimental design of a separate experiment with PDX-2 experiment. In this study, mice injected with BPDCN-3 ($1 \times 10^6$ tumor cells/mouse) received similar treatment, this time in 5 groups (3 UCART123 doses; n = 9–10 mice/group) that was initiated on day 14 after injection of tumor cells, when tumor burden was low (0% circulating BPDCN, 0.9–1.1% engraftment in the bone marrow). **f–h** UCART123 extended survival (**f**) and reduced or eliminated circulating BPDCN cells (**g**; no engraftment was seen in cohorts that received 3 or $10 \times 10^6$ UCART123) when therapy initiated at a low tumor burden. IFNγ levels in peripheral blood were measured 2 days following T cell injection (**h**) and were considerably lower than in the 1st study. MST = median survival time (in days). The mean ± SD value is presented. For **b**, **d**, **f**, **h**, Compare to vehicle group: Significance was determined using unpaired two-tailed t-test annotated as "NS" (not significant, $p \geq 0.05$), **$P \leq 0.01$, ***$P \leq 0.001$, ****$P < 0.0001$. Source data are provided as a Source data file.

**Molecular analysis of CD123 loss following UCART123 treatment.** To understand the molecular basis for loss of CD123 surface expression in PDX-3 BPDCN cells, we isolated RNA from CD123+ blasts from 2 of the vehicle-treated mice and CD123- blasts from 4 of the UCART123 treated mice, and performed RT-PCR and total RNA-sequencing. The cells from all samples were hCD45+ and hCD56+, indicating they were leukemia cells. These analyses detected the presence of full-length

transcripts containing exons 2–12 in both CD123+ samples (#1 and #2 in Fig. 6a). In 2 of the 4 CD123- samples, CD123 transcripts were completely absent, as were transcripts of neighboring genes (#3 and #9 in Fig. 6a). RNA-sequencing reads aligned to Genome Browser tracks for CD123 and housekeeping gene GPI showed no reads present for CD123 but reads present for GPI in the 2 samples with CD123 loss (Supplementary Fig. 13a). The Array-based Comparative Genomic Hybridization (aCGH)

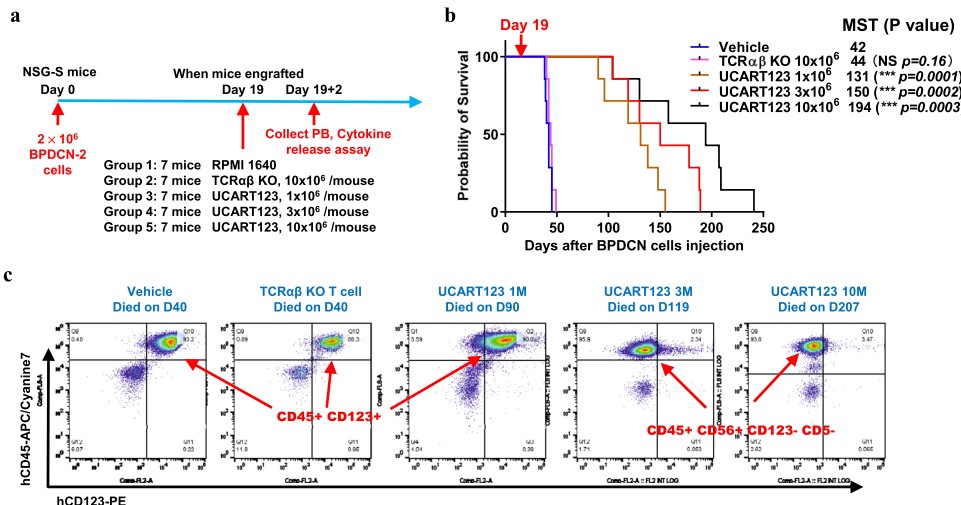

**Fig. 5 Loss of CD123 leads to escape from UCART123 therapy and causes early relapses. a** NSGS mice were injected with cells from a primary BPDCN sample (BPDCN-2) to create a xenograft experiment (PDX-3). After documenting engraftment on day 19 (0% circulating BPDCN, 13.9% and 33.8% engraftment in bone marrow from 2 mice), mice were randomized into 5 treatment groups (n = 7 mice/group) and were treated as follows: vehicle; $10 \times 10^6$ TCRαβ KO T cells; $1 \times 10^6$ UCART123 cells; $3 \times 10^6$ UCART123 cells; or $10 \times 10^6$ UCART123 cells. **b** Survival of mice in the treatment groups was estimated by the Kaplan–Meier method. Red arrow indicates start of treatment (Day 19). MST = median survival time (in days). Significance was determined using unpaired two-tailed t-test annotated as "NS" (not significant, $p \geq 0.05$), ***$P \leq 0.001$. **c** Expression of CD123 on tumor cells isolated from bone marrow of control or UCART123-treated mice at the time they were sacrificed due to excessive tumor burden. Tumor cells from control mice co-expressed CD123 and CD56, while in UCART123-treated mice, tumor cells lost CD123 expression (CD123⁻) but remained CD56⁺. M = million. Source data are provided as a Source data file.

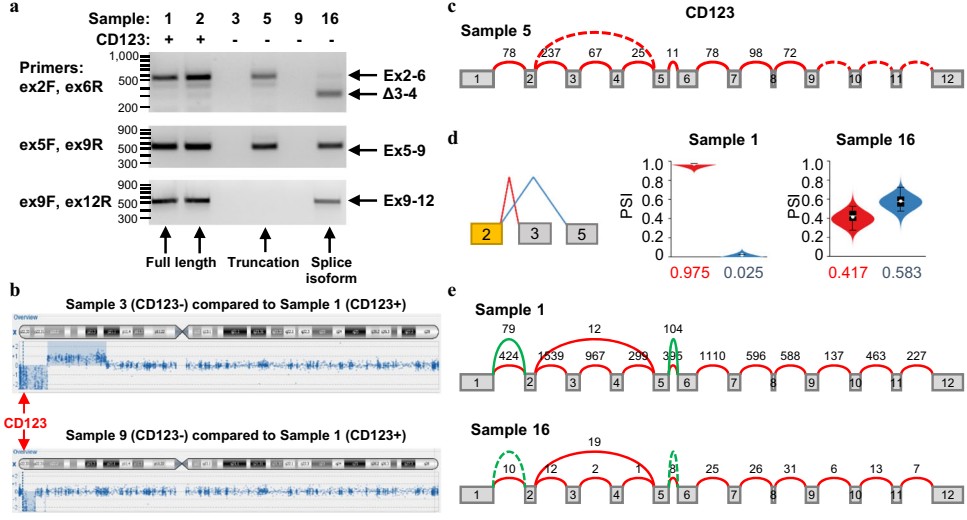

**Fig. 6 Molecular analysis of CD123 loss following UCART123 treatment. a** RNA was isolated from 2 CD123⁺ samples from the vehicle-treated group (samples 1 and 2) and 4 CD123⁻ samples from the UCART123-treated group (samples 3, 5, 9 and 16). RT-PCR performed with primer pairs spanning the entire CD123 transcript detected full-length CD123 RNA (samples 1 and 2), truncated C123 RNA (sample 5), spliced CD123 isoforms (sample 16), or absence of CD123 mRNA (samples 3 and 9). Each RT-PCR experiment was repeated twice, on different dates, with essentially identical results. 100 bp DNA Ladders were used as molecular weight markers. Uncropped and unprocessed scans can be found in the Source Data file. **b** aGCH analysis performed on samples 3 (upper) and 9 (lower) showed a large genetic deletion on chromosome X, compared to CD123⁺ samples, corresponding to the location of CD123. **c** A Majiq-generated splice graph shows no exon junction reads for exons 9–12 of CD123 in sample 5. **d** Majiq-generated violin plots show the predominant splicing of exon 2 to exon 3 (red) in control (CD123 + ) samples but an increase in skipping from exon 2 to exon 5 (blue) in sample 16. The 95% confidence interval in this context is defined as follows: given observed reads in both experiments there is a 95% posterior probability that there is a change of 20% or more in percent-spliced-in (PSI). **e** A Majiq-generated splice graph shows increased skipping of exons 3–4 in sample 16 (lower panel) compared to control CD123⁺ samples (upper panel). Source data are provided as a Source data file.

results showed that, compared to sample 1 (CD123⁺), samples 3 and 9 (CD123⁻) had large regional deletions on chromosome X, which includes the CD123 gene (Fig. 6b). In another CD123⁻ sample (sample 5), the splicing analysis algorithm MAJIQ[19,20] detected CD123 transcripts containing only exons 2–9, indicating

premature transcription termination. Interestingly, if translated, this truncated transcript would produce a protein isoform lacking the transmembrane domain in exon 10 (Fig. 6c and Supplementary Fig 13b). Finally, MAJIQ also revealed canonical splicing of exon 2 to exon 3 in all CD123⁺ samples but a sharp increase in

skipping from exon 2 to exon 5 in sample 16 (Fig. 6d, e). This exon-skipping event preserves the open-reading frame and yields the previously reported transcript variant 2 (NM_001267713). Per UniProt[21] the resultant protein will retain the ligand-binding domain but lacks several glycosylation sites and two beta sheets in the extracellular domain, potentially compromising recognition by UCART123. The aCGH and FISH results further showed that this patient sample harbored TP53 deletion (Supplementary Fig. 14). Gene set enrichment analysis showed that CD123 loss was associated with reduction of expression of genes in SRC/STAT3 pathway, consistent with inability of CD123 receptor to transmit signals from IL-3 ligand (Supplementary Fig. 15). These results indicate that the BPDCN relapses could develop CD123 loss through different genetic mechanisms.

## Discussion

Chimeric antigen receptor (CAR) gene therapy from autologous T cells is a breakthrough technology in treating refractory hematologic malignancies[22,23]. However, there are limitations, including inability to manufacture CAR-T cells from the patient's T cells, or disease progression prior to completion of cell production and release of the engineered autologous product[24,25]. T cell dysfunction is known to occur in cancer patients, and several groups have recently described differences in CAR-T cells generated from chronic lymphocytic leukemia (CLL) patients compared with those from a healthy donor[26,27]. This is thought to contribute to the low response rates observed in this disease group. Allogeneic gene-edited CART products, which do not require human leukocyte antigen (HLA) matching, and minimize the risk of graft versus host disease, offer the potential advantages of healthy donor-derived CAR-T cells as an "off the shelf" product, which overcomes the manufacturing difficulties of producing CAR-T cells for each individual patient[17].

The cell surface antigen CD123 is highly expressed and is a hallmark of BPDCN, highlighting the role of CD123 as a therapeutic target in BPDCN. In this study, the antitumor activity of allogeneic off-the-shelf CD123 CAR T cells (UCART123) was demonstrated by in vitro and in vivo assays, with both a BPDCN cell line and primary BPDCN samples. Albeit no correlation between CD123 expression in primary BPDCN and in vitro cell lysis could be demonstrated (not shown), the specificity was confirmed by the lack of cytotoxic activity of UCART123 against CD123-negative Daudi cells and a trend towards increasing degranulation, cytotoxic activity and IFNγ release by CART cells against AML cell lines with higher CD123 levels (Supplementary Fig. 3). Treatment with UCART123 cells markedly prolonged survival of the UCART123 treated PDX-mice and eliminated detectable BPDCN cells. Of importance, we have demonstrated long-term persistence of UCART123 in bone marrow and spleen of mice injected with PDX-1 and PDX-3 models (by flow or ddPCR, Supplementary Fig. 8 and Supplementary Table 1). It should be noted that the persistence in humans is expected to be limited by the rejection of the allogenic product upon recovery from lymphodepletion, which cannot be tested in the immunodeficient mouse models. Encouragingly, according to recent reports in relapsed/refractory B-ALL treated with UCART19[28] and in NHL patients treated with ALLO-501[29], both products bearing *TRAC* knock-out generated using TALEN® technology, showed high response rates and allogenic CAR-T cell expansion, supporting the notion that even limited CAR-T cell in vivo persistence is sufficient to produce meaningful responses.

Cytokine release syndrome (CRS), resulting from rapid immune activation induced by CAR-T cells, is the most frequent treatment-related toxicity seen in clinical trials[30,31]. Similar to other immunotherapy products, UCART123 can potentially induce cytokine release syndrome or tumor lysis syndrome. In the PDX-2 experiment, administration of UCART123 to mice with high tumor burden caused a burst of IFNγ secretion and rapid demise of animals, possibly from cytokine release and/or tumor lysis syndromes. This was supported by a second experiment whereby administration of UCART123 cells in mice harboring the same PDX at a lower tumor burden, was associated with drastically lower IFNγ levels and translated into long-term survival of UCART123-treated animals. The lentiviral construct of UCART123 allows expression of the safety switch RQR8, permitting the cells to be eliminated by rituximab (RTX) through Complement-dependent Cytotoxicity (CDC) and Antibody Dependent Cell-mediated Cytotoxicity (ADCC) (Supplementary Fig. 1). Our in vivo data confirm that RQR8 targeting by rituximab leads to CAR-T elimination associated with abrogation of cytokine production (Supplementary Fig. 10). The safety switch should allow elimination of UCART cells in the setting of severe CRS in patients, as confirmed by the efficacy of Rituximab in vitro (Supplementary Fig. 1) and by the efficient depletion in vivo of RQR8-expressing T-cells by rituximab reengineered to mouse IgG2a, reported by Pule et al.[18].

With respect to on-target off-tumor toxicity of CD123-targeting CART, we have shown minimal toxicity of UCART123 against normal hematopoietic cells in vitro (Supplementary Fig. 6). While the toxicity against endothelial and other normal cells was not addressed in this study, preliminary results from autologous CD123 CART clinical trial at the City of Hope[32] indicated lack of bone marrow or other organ toxicity, and no evidence for capillary leak syndrome. Similarly, in an ongoing clinical trial with anti-CD123 antibody-drug conjugate IMGN632, no endothelial or marrow toxicity was seen[33,34] in AML or BPDCN patients, supporting acceptable safety targeting CD123. Treatment with CAR T-cells targeting the CD19 antigen (CART-19) has yielded very high response rates in patients with B-ALL. However, 30–60% of patients relapse after CAR treatment, and among those, 10–20% are CD19-negative relapses[35]. The main underlying mechanism is the selection for preexisting alternatively spliced CD19 isoforms with the compromised CART-19 epitope[36–38]. In one (PDX-3) out of three PDX studies, mice treated with UCART123 experienced CD123-negative relapse. Unexpectedly, we identified not a single, but three diverse mechanisms of CD123 loss amongst mice that relapsed with CD123- disease: genomic loss of CD123 DNA; a truncated transcript containing only exons 2–9; and predominant alternative splicing (skipping from exon 2 to exon 5) which is reminiscent of CD19 epitope loss in B-cell acute lymphoblastic leukemia[36]. The most plausible explanation is the selection of a small pre-existing population of clones lacking CD123, that were enriched upon the selective pressure of UCART123 therapy. It is worth noting that this PDX experiment was derived from a patient whose tumor harbored a TP53 mutation, which could have contributed to genomic instability observed here in different mice engrafted with the same tumor cells. To our knowledge, this is the first reported observation of genomic CD123 loss in an animal study causing early relapses post CD123 targeted CART therapy. While relapses on the tagraxofusp or IMGN632 clinical trials are generally not associated with CD123 loss in BPDCN[34,39], this observation could be highly relevant for monitoring strategies targeting CD123, including autologous CART[40,41], CD123-ADCs[42], BiTes[43,44], and DARTs[45,46].

In summary, allogeneic UCART123 therapy resulted in eradication of BPDCN in vitro and in long-term disease-free survival in a subset of primary BPDCN PDX experiments. Our findings in one of the PDX experiments cautioned that CD123 loss could compromise efficacy of this approach with competitive survival of CD123 negative clone(s). Still, these results provide preclinical

proof-of-principle that allogeneic UCART123 cells have potent anti-BPDCN activity.

## Methods

**Study approval and ethics statements.** Primary BPDCN cells and AML cells were collected from patients who had consented to research protocols approved by the Institutional Review Board at The University of Texas MD Anderson Cancer Center (PA14-0157, A Study to Evaluate Patient Samples with Blastic Plasma Cytoid Dendritic Cell Neoplasm). This animal study was performing under Animal Care and Use Protocol (Immunotherapy approaches in leukemia, 00001446-RN00). The production of R&D grade batches of UCART123 was done using commercially available NHPBMC cells ordered from ALLCELLS or HEMACARE. These are normal human peripheral blood mononuclear cells derived from an apheresis collection. Donors have been screened for viral status and the collection protocols and donor-informed consent are approved by an Institutional Review Board (IRB), with strict oversight. HIPAA compliance and approved protocols are also followed. The commercially available NHPBMC cells used to generate the UCART cells described in the manuscript are for Research Purposes Only and were used according to local licenses and regulations (French Ministry of Higher Education, Research and Innovation – French Biotechnology Council).

**Cell lines.** The BPDCN CAL-1 cells[47] were obtained from Takahiro Maeda (Nagasaki University). MOLM13 acute myeloid leukemia (AML) and Jurkat T cell leukemia cells were obtained from ATCC (Manassas, VA) or Deutsche Sammlung von Mikroorganismen und Zellkulturen (Braunschweig, Germany). All cells were validated by short-tandem repeat profiling prior to their use in these experiments, and underwent mycoplasma testing every 6 months. All three cell types were grown in RPMI 1640 medium supplemented with 10% heat-inactivated fetal bovine serum (FBS). The cells were cultured at 37 °C with a 5% $CO_2$-in-air atmosphere.

**Patient samples.** Primary BPDCN cells and AML cells were collected from patients who had consented to research protocols approved by the Institutional Review Board at The University of Texas MD Anderson Cancer Center for analysis of hematologic malignancies (PA14-0157, A Study to Evaluate Patient Samples with Blastic Plasma Cytoid Dendritic Cell Neoplasm). Bone marrow aspirate mononuclear cells from patients with BPDCN were purified by Ficoll density centrifugation using standard procedures. Lymphocytes were isolated by using Lymphocyte Separation Medium. Cells were frozen in FBS and dimethyl sulfoxide and stored in liquid nitrogen until use.

**CAR construct.** The CAR construct used is described in patent EP3119807B1 ("CD123 specific chimeric antigen receptors for cancer immunotherapy"). We used the scFv derived from the monoclonal antibody Klon43 fused to CD8a hinge and transmembrane domains, the 4-1BB co-stimulatory domain, and the CD3zeta signaling domain.

**UCART123 production.** R&D grade UCART123 cells were produced by Cellectis using a large-scale manufacturing process. UCART123 cells were derived from a single donor. Two large scale batches (derived from the same donor) were used for in vitro testing, while a single batch was used in all the in vivo experiments. For the data presented in Supplementary Fig 3, three small scale batches were used, from three independent donors. Non-transduced TCRαβ-deficient T-cells (TCRαβ KO T-cells) were produced from the same donor and in the same conditions, to be used as control cells. Briefly, PBMCs from healthy donors were thawed and T-cells were activated using CD3/CD28 magnetic beads the day after thawing. Cultures were performed at 37 °C in 5% $CO_2$ in X-vivo15 media, supplemented with 5% Human AB Serum and 20 ng/ml of human IL-2. Cells were transduced with a recombinant lentiviral vector encoding the CD123-targeting CAR, at MOI 5, three days after activation. Two days later the cells were transfected with TALEN® mRNA (using the Pulse Agile® electroporation system) to knockout the TRAC gene and disrupt TCRαβ expression. Cells are then expanded at 37 °C in 5% $CO_2$ within a WAVE bioreactor Cellbag of 2 L or 10 L on the Xuri™ Cell Expansion System W25 for 10 days. At the end of the expansion phase, TCRαβ negative cells are isolated by negative selection using a TCRαβ biotin and anti-biotin magnetic bead system from Miltenyi, using the CliniMACS device. The purified TCRαβ- cell population was then resuspended in cryopreservation media, aliquoted in cryovials and stored at below −135 °C.

**Cytotoxicity assay.** The cytotoxic activity of UCART123 towards the BPDCN cells (CAL-1 cell line or primary samples) was assessed by measuring the survival of tumor cells when co-cultured with UCART123 cells. The CD123+ MOLM13 cell line was used as a positive control. MOLM13 and CAL-1 cells were stained with carboxyfluorescein succinimidyl ester, co-incubated with UCART123 cells at a 1:1, 3:1, or 10:1 effector-to-target (E:T) ratio for 16 h, and the viability of the target cells was determined by flow cytometry using eFluor 780 fixable viability dye. Flow cytometry data were analyzed using by Flowjo V10. The viability of UCART123-treated cells was compared to that of samples co-cultured with non-transduced

TCRαβ-deficient (TCRαβ KO) T cells to determine the percentage of specific cell lysis.

%Lysis = (1−viability of UCART123-treated cells / viability of TCRαβ KO T cells-treated cells) × 100%

**T cell degranulation assay.** UCART123 cells were co-incubated with BPDCN samples, or with CD123- Jurkat cells as a negative control, at a ratio of 1:1 with an anti-CD107α antibody for 6 h. Co-activator reagents (monensin and anti-CD49d and antiCD28 antibodies) were added to the mixture. UCART123 cells incubated with phorbol myristate acetate (PMA)/ionomycin were used as a positive control of T cell activation. Cells were analyzed for CD107α signal by flow cytometry, gating on viable and CD8+ cells. Flow cytometry data were analyzed using by Flowjo V10.

**IFNγ release assay.** The release of IFNγ by activated T cells was analyzed by using a BioLegend LEGENDplex assay. Briefly, TCRαβ KO T cells or UCART123 cells were co-cultured for 25 h at a 1:1 (E:T) ratio with CD123- (Jurkat), CD123+ (MOLM13 or CAL-1), or primary BPDCN cells. The quantity of IFNγ released into the culture supernatants was measured. PMA/ionomycin was used as a positive control. The same BioLegend LEGENDplex assay was used to evaluate IFNγ and IL-2 release levels in the PDX experiments.

**Animals.** The animal study was performing under Animal Care and Use Protocol at The University of Texas MD Anderson Cancer Center (Immunotherapy approaches in leukemia, 00001446-RN00). NSGS mice (NOD-scid IL2Rgnull-3/GM/SF, NSG-SGM3) and NOD scid gamma (NSG) mice were used in this study. The mice were female, 8–12 weeks old. The standard area cages were used with a maximum of 5 mice per cage (from the same group) according to MD Anderson internal standard operating procedures. The mice were given bactrim-containing food and acidified water. Animals were identified with different markers on tail. Animals were maintained in rooms under controlled conditions of temperature (22 ± 2 °C), humidity (55 ± 10%), photoperiod (12 h light/12 h dark) and air exchange. Animals were maintained in SPF conditions. Room temperature and humidity were continuously monitored. Mice were monitored at least once daily, and up to twice daily to follow eventual morbid signs: hunched posture, rapid weight loss (more than 20% of body weight from baseline), ruffled fur, impaired mobility (difficulty reaching food and water), limb paresis etc. Animals that exhibited one of these morbid signs were immediately euthanized by compressed $CO_2$.

**Patient-derived xenografts.** After being quickly thawed in a 37 °C water bath, cryopreserved primary BPDCN cells were resuspended in 10 mL of MEM, Alpha 1X (Corning) supplemented with 20% FBS (Invitrogen), and centrifuged at 300 g for 5 min. Cells were resuspended in mixture of 10 mL of MEM, Alpha 1X containing 20% FBS, 0.1 mg/ml heparin, 10 units/mL DNase (Thermo Scientific), and 10 mM MgSO4 (Sigma). Cells were incubated for 15 min in 37 °C incubator in the above mixture and centrifuged at 300 g for 5 min. After centrifugation, BPDCN cells were resuspended in the injection medium (serum-free RPMI 1640 medium) at a concentration of $2 \times 10^6$ BPDCN cells/100 μL. Cells were kept on ice until injection into immunodeficient NSG or NSGS mice via the tail vein (100 μL/mouse). Mice were monitored weekly for evidence of human BPDCN in the peripheral blood. When circulating human CD56+CD123+ cells (i.e., BPDCN cells) were detected, mice were randomized into treatment groups ranging from 7 to 12 mice per group. Treatments comprised a single tail vein injection of vehicle; $10 \times 10^6$ non-transduced TCRαβ KO T cells; $1 \times 10^6$ UCART123 cells; $3 \times 10^6$ UCART123 cells; or $10 \times 10^6$ UCART123 cells. Rituximab (Genentech, USA) was given i.p. at 10 mg/kg dose for 5 days.

**Human tumor cell analysis by flow cytometry.** The presence of human tumor cells in viable (DAPI−) cells isolated from mouse blood or spleen was identified by analyzing cell surface expression of hCD123 and hCD56. The percentage of engraftment was calculated as indicated below:

Engraftment % = CD123+CD56+ cells/DAPI− cells × 100%

For the analysis of CD marker expression, the red blood cells were subjected to lysis in ammonium chloride solution. Samples were incubated with a mixed solution of anti-CD123 antibody and anti-CD56 antibody. After DAPI staining, viable cells (DAPI-) were analyzed with a Gallios flow cytometer (Beckman Coulter). The acquisition was stopped after 5,000 DAPI- viable cells (if achievable) for each sample. All the events were saved during acquisition. Flow cytometry data were analyzed using Flowjo V10.

**Detection of UCART123 cells by flow cytometry.** The presence of UCART123 cells in blood was analyzed using cell surface immunostaining for hCD5 (human T cells, Cat#555352, BD Pharmingen) and for anti-CD123 CAR, using a synthetic protein comprising the extracellular portion of hCD123, fused to a mouse Fc (custom made, LakePharma, Inc). The peripheral blood, spleen, and bone marrow were collected from each mouse and were subjected to lysis in ammonium chloride solution. The red blood cells were subjected to lysis in ammonium chloride solution. Samples were incubated with Fc block master mix at room temperature for 10 min. Then samples were incubated with the CD123-Fc protein (150 ng protein/

300,000 cells) for 15 min at 4 °C. Cells were washed and then resuspended in 50 µL of hCD5 and anti-Fc secondary antibody (Cat#115-115-164, Jackson ImmunoResearch). After DAPI staining, viable cells (DAPI⁻) were analyzed by flow cytometry. The acquisition was stopped after 5000 DAPI⁻ viable cells (if achievable) were collected for each sample. All the events were saved during acquisition. Where indicated, cells were stained with hCD45 (Cat#304014, BioLegend), mCD45 (Cat#103111, BioLegend), hCD5, anti-TCRαβ (Cat#130-098-782, Miltenyi Biotec), and anti-CAR123-Fc antibody followed by anti-Fc secondary antibody. Flow cytometry data were analyzed using by Flowjo V10.

**Detection of CD123 expression level in AML, AML remission, and BPDCN patient samples.** Multiparameter flow cytometry (MFC) analysis of CD123 expression in neoplastic cells of patients with BPDCN or AML or AML remission was performed on FACS Canto II analyzers (BD Biosciences, Mountain View, CA) using bone marrow aspirate material collected in EDTA. Details of the MFC panel and gating strategies used at MD Anderson Cancer Center were described by our group previously[48,49]. The blast gate was defined on the basis of CD45dim expression (CD34 for patients with AML remission) and side-scatter characteristics and quantified as a percentage of total gated events. CD123 expression was measured using an allophycocyanin-conjugated anti-CD123 (IL-3 receptor α chain) antibody (clone 7G3; BD Pharmingen, BD Biosciences) according to the manufacturer's recommendation. At least 100,000 events per sample were acquired. Data were analyzed using FCS Express software (De Novo Software, Los Angeles, CA). CD123 expression was reported as percentage of positive blasts and as mean fluorescence intensity (MFI) on leukemic blasts.

**Digital droplet PCR.** UCART123 transcripts in spleen were evaluated with digital droplet PCR (ddPCR) at the timepoint of euthanasia due to sickness or disease progression. RNA was extracted by Qiagen RNeasy Kit and 25–50 ng of tRNA was converted into cDNA with SuperScript VILO (Thermo Fisher Scientific). Ten million droplets were generated from cDNA samples (step1) and single molecule in droplets were amplified by PCR reaction (step2) followed by counting the absolute number of fluorescent droplets (step3). Step 1 and step 3 were performed and analyzed with RainDrop and RainDrop Analyst II software (RainDance Technologies/Bio-Rad Technologies). Copy numbers of UCART123 transcript relative to $1.0 \times 10^4$ ABL1 transcripts (reference) were calculated.

**aGCH assay.** The genomic DNA was extracted from CD123- cells from UCART123 treated mice in PDX-3 experiment. The quality of genomic DNA was tested by NanoDrop 2000 (Thermo Scientific). 1.0 µg genomic DNA from CD123 + cells from vehicle-treated mice, which is used as reference, was labeled with Cy3 and Cy5 fluorescent dyes, respectively, by using CytoSure™ HT Genomic DNA Labelling Kit (Cat# 500040, Oxford Gene Technology, UK). Labeled DNA was further purified with Clean-up columns (Cat# 500020, Oxford Gene Technology, UK) to remove unincorporated fluorescent dyes. Labeled DNA was measured the absorbance at 260 nm, 55 nm and 650 nm by NanoDrop 2000. Equal amount of labeled DNA from CD123- cells and CD123 + cells were mixed in hybridization buffer, loaded on 8x60K CytoSure ISCA + SNP array (Cat#20052, Oxford Gene Technology, UK) and hybridize at 65 °C for 22 h in the hybridization oven. After post washing, the array slide was scanned by InnoScan 710 Microarray Scanner (Innopsys, France) and further analyzed using CytoSure Interpret software version 4.8.32 (Oxford Gene Technology, UK).

**RT-PCR.** RNA was isolated using Qiagen RNeasy Kit and used for RT-PCR or RNA Sequencing. For RT-PCR, cDNAs were prepared with random hexamers using the High Capacity cDNA RT Kit (Life Technologies). CD123 was amplified by semi-quantitative PCR for 30 cycles with primer pairs spanning the coding exons (exon 2F, exon 6R; exon 5F, exon 9R; exon 9F, exon 12R, Supplementary Table 2). Amplicons were visualized on a 1.5% agarose gel. Band identities were confirmed by gel extraction and Sanger Sequencing.

**RNA sequencing analysis.** Prior to sequencing, RNA integrity and concentration were determined using Eukaryote Total RNA Nano assay on BioAnalyzer. RNA-seq was performed on 10 ng to 1 µg of total RNA according to the GeneWiz Illumina Hi-seq protocol for poly(A) selected samples (2 × 150 bp pair-end sequencing, 350 M raw reads per lane). Fastq files of RNA-seq obtained from GeneWiz were mapped using STAR aligner. STAR was run with the option 'alignSJoverhangMin 8'. We generated STAR genome reference based on the hg19 build. Alignments were then quantified for each mRNA transcript using HTSeq with the Ensembl-based GFF file and with the option '-m intersection-strict'. Normalization of the raw reads was performed using the trimmed mean of M-values (TMM). In order to generate splice-graphs, the MAJIQ tool (version 1.03) was used. MAJIQ was run on the Ensembl-based GFF annotations, disallowing de novo calls. The VIOLA visualization package was used to plot the resulting splice graph, the alternative splicing variants and the violin plots representing the PSI values.

**Pathway analysis.** Pathway analysis was carried out using the gene set enrichment analysis (GSEA) method[50]. Gene level RNASeq data were used as input. Samples were divided into CD123 + and CD123- classes. GSEA-R program was used to search against the entire gene set collection of Molecular Signatures Database v6.1.

**Statistical analysis.** Statistical analyses were performed using Excel 2013, GraphPad Prism V8. Statistical tests was done using the two-tailed unpaired t-test. The statistical test used for each Figure is described in the corresponding Figure legend.

**Reporting summary.** Further information on research design is available in the Nature Research Reporting Summary linked to this article.

## Data availability

The RNA sequencing data discussed in this publication have been deposited in NCBI's Gene Expression Omnibus and are accessible through GEO Series accession number GSE198471. The remaining data are available within the Article, Supplementary Information or Source Data file. Source data are provided with this paper.

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

## Acknowledgements

This work was in part supported by Cellectis. This study used the Flow Cytometry and Cellular Imaging Core Facility at The University of Texas MD Anderson Cancer Center and was supported by the National Institutes of Health/NCI under award number P30CA016672.

## Author contributions

T.C. and M.K. conceived the study. T.C., A.G., R.G., M.G., J.S., and M.K. designed and conducted the main analyses and interpreted the results. T.C., M.Z., A.S., C.W., M.S., A.C., L.H., Q.Z., V.K., H.M., C.L., X.L., S.K., G.A., and S.C. conducted the laboratory experiments. T.C., A.G., K.B., A.S., A.N., D.T., Q.Y., R.G., S.F., S.K., J.G., G.T., X.S., G.A., S.C., S.N., A.L., H.K., M.G., N.P., J.S., and A.T. contributed data and/or analysis advice to the study. T.C., A.G., K.B., R.G., J.S., and M.K. wrote the manuscript, with contributions and review by all other authors.

## Competing interests

A.G., R.G., S.F., and J.S. are Cellectis employees. S.S.N., M.L.G., N.P., and M.K. received research funding from Cellectis. The remaining authors declare no competing interests.
