## [Peer Review File · Nature Communications]

Reviewers' Comments:

Reviewer #2:

Remarks to the Author:

The authors generated allogeneic T cells expressing a second generation anti-CD123 chimeric antigen receptor and tested them against the rare leukemia BPDCN that expresses high levels of CD123.

What are the noteworthy results?

Patients with BPDCN need new treatments since the prognosis is very low. CD123 CART cells have already been tested in detail preclinically in AML and also BPDCN and they are already evaluated in clinical trials. The novelty of this study is the use of an allogenic CAR T cells with TALEN nuclease edition and in providing a mechanistic study about the loss of CD123 observed in one of three BPDCN PDX studied.

However, no toxicity study has been reported here whereas it seems to be mandatory in the context of allogenic cells and since CD123 is expressed on some hematopoietic stem cells, endothelial cells, monocytes and basophils. Moreover, the assessment of the persistence of allogenic CAR T without TCR is a major issue since the community of hematologist is anxious to know if these CARs without functional TCR will be able to persist in vivo and have long-term action, which is a crucial point of CAR therapies, now known.

Does the work support the conclusions and claims, or is additional evidence needed?

The report is well written and experiments well controlled.

However a study about potential on tumor/off target toxicity should be done, even if UCART123 have a safety switch, mainly because its functionality has not been demonstrated in the report and because a patient died with this/or another version of UCART123. Moreover, testing of in vivo persistence would be of great interest.

Are there any flaws in the data analysis, interpretation and conclusions? - Do these prohibit publication or require revision?

The lentiviral construct contains the safety switch RQR8. The functionality of this system is not discussed in vitro nor in vivo.

Figure 2a-b : All of the 8 BPDCN patients express similar and high level of CD123, however UCART123 demonstrated cytotoxicity against 5 of the 8 samples, what are the hypothesis of the authors about this lack of toxicity on CD123+ blasts ? It may be interesting to put CD123 MFI of CAL-1 cells on the graph 2a to compare with BPDCN patients.

Figure 1b and 2b : The authors presented specific lysis of UCART123, I think it could be interesting to present cell death after co-culture with TCR $\alpha\beta$ KO T cells in comparison with co-culture UCART123 allowing to show potential alloreactivity as it is done for degranulation experiments and IFN- γ secretion.

Figure 3 : The level of BPDCN engraftment in blood before CAR injection might be detailed to control the level of the different groups. Tumor burden might be given in absolute values (blasts/ μ L) to compare precisely evolution before and after CAR injection.

Extended data figure 3A mouse of the 10×10^6 UCART123 treatment group died, the authors did not detect BPDCN cells in the BM so they conclude that the death is not due to BPDCN progression. Do they study other organs such as spleen or lungs ?

Figure 4, PDX models : it is not explained why do authors use sometimes NSG mice and sometimes MSG-S mice especially for BPDCN-3 cells. Authors observed CRS syndrome in NSG mice and not in MSG-S mice, the reason of that observation might be suggested. Concerning the CRS observed, as the construct has the RQR8 safety switch ? if yes, the use of rituximab to diminish the CRS might be tested

- Is the methodology sound? Does the work meet the expected standards in your field?

It is ok.

- Is there enough detail provided in the methods for the work to be reproduced?

The methods are well written, detailed and clear.

Reviewer #3:

Remarks to the Author:

The authors characterize the performance of their original healthy donor-derived CAR T cells targeting CD123 on BPDCN, with in vitro killing assay and PDX models. The CAR T cells were constructed with the TCRA constant gene disrupted by the TALEN technology. Multiple issues need to be overcome for the success of allogeneic CAR T cell therapy directing CD123 on AML and BPDCN; 1) anti-tumor cell efficacy, 2) control of GVHD, 3) measures against allogeneic immune-related eradication of CAR T cells, and 4) intentional eradication of CAR T cells to protect normal hematopoietic cells. The authors' CAR T cells are constructed aiming at above 1), 2), and 4). In the current manuscript, however, only the anti-tumor cell efficacy is evaluated, although TCR gene disruption to eventually aiming at avoiding GVHD may have helped evaluating the anti-tumor activity in the PDX models.

Major

1. There is an obvious limitation in the manuscript as indicated above. The authors should clearly define that their purpose is to show the aspect of anti-tumor activity of their CD123 CAR T cells, discriminating from other issues. Then, they need to describe comparison between the preceding anti-CD123 CAR T cells and theirs simply in terms of the efficacy. To this end, it is important to describe the anti-tumor activity and the expression levels of CD123. The in vitro data do not show the correlation between the efficacy and the expression levels of CD123 among BPDCN samples. It may be considered to incorporate AML cases with demonstration of CD123 levels.
2. The authors should describe whether they found a significant effect of TCR depletion in the PDX experiment. Because the PDX models are important for the evaluation of many CAR T cells, it may be helpful if they compare UCART123 with the version holding TCR.
3. It is important to clarify the frequency of the downregulation of CD123 by genetic events as the mechanism of the acquisition of resistance, particularly if there is a possibility of differences in the frequencies depending on the anti-CD123 treatment modalities including SL-401.

Minor

1. The major evaluation by PDX models were performed with 3 patients-derived samples. Whereas the authors described PDX model 1, 2, and 3, they used samples from patients #1-3 for No1, #3 for No2, and #2 for No3 experiment with the completely the same or very similar experimental designs to evaluate the reproducibility. It should not be designated as a "model" but just "experiment."

Reviewer #4:

Remarks to the Author:

In the present manuscript the authors present preclinical data for an off-the shelf CAR-T product directed against CD123. Overall, the manuscript is very well written and the data presented in a clear way. However, preclinical data for CAR-T targeting CD123 for treatment of AML or BPDCN have been previously presented by other groups. Hence, the manuscript lacks novelty in regard of the therapeutic approach. The emergence of a CD123 escape mutant in the in vivo model is interesting for the clinical translation of the approach and should be further investigated. Is the emergence of escape mutants specific to the applied CAR-T product or maybe related to the used CD123 binder used for the CAR construct?

The transnational value of the presented data remains unclear, as UCART123 is/was already under clinical investigation for BPDCN. A study hold was put on a study after death of the first patient treated with UCART123. Public information on the current status of the study is lacking. In the present manuscript it is described that UCART123 harbors as safety switch (Rituximab target epitope). Is the safety switch non-function or why did it not work in the clinic situation? The manuscripts lack any data how to tackle such acute toxicities and also lack any data/explanation how long-term toxicities against CD123+ expressing hematopoietic progenitor cells should be mitigated. Adding such data would enhance the scientific value of the manuscript as in the present manuscript the pathway towards further clinical translation of the approach is unclear.

In Figure 1 and Figure 2, CD107a up-regulation is shown for CD8+ UCART123 cells, but data is lacking for CD4+. Are CD4+ cells contributing to target cell lysis in UCART123 products as shown by other groups for their CD123+ CAR-T? What is the CD4/CD8 composition of a the tested

UCART123 batches?

In the second in vivo model experimental mice treated with UCART123 died early likely due to CRS. Repeating the same experiment with a lower number of leukemic cells and earlier on-set of treatment resulted in long-term survival of treated mice. However, using the same number of leukemic cells and starting treatment at a similar time point as in the experiment shown in Fig. 4A did not result in CRS. The reviewer concludes that severe CRS is obviously very much dependent on the leukemia kinetic in vivo and maybe specific interaction between the UCART123 and the leukemic cells. However, this problem will be very likely also encountered in the clinical setting. The authors should at least provide an explanation in the discussion section how they envisage to encounter acute toxicities as already observed with UCART123 in the clinic.

NCOMMS-20-29505-T entitled “Targeting CD123 in Blastic Plasmacytoid Dendritic Cell Neoplasm using Allogeneic Anti-CD123 CAR T Cells”

We thank the reviewers for taking the time to carefully and critically review our manuscript. We believe we have addressed each of the concerns. In addition, all references have been updated. Please find below point by point our responses. We are hoping that the responses to the Reviewers’ criticism has significantly improved the impact and readability of the manuscript.

RESPONSE TO REVIEWER COMMENTS

Reviewer #2 (Remarks to the Author):

The authors generated allogeneic T cells expressing a second generation anti-CD123 chimeric antigen receptor and tested them against the rare leukemia BPDCN that expresses high levels of CD123.

What are the noteworthy results?

Patients with BPDCN need new treatments since the prognosis is very low. CD123 CART cells have already been tested in detail preclinically in AML and also BPDCN and they are already evaluated in clinical trials. The novelty of this study is the use of an allogenic CAR T cells with TALEN nuclease edition and in providing a mechanistic study about the loss of CD123 observed in one of three BPDCN PDX studied.

However, no toxicity study has been reported here whereas it seems to be mandatory in the context of allogenic cells and since CD123 is expressed on some hematopoietic stem cells, endothelial cells, monocytes and basophils. Moreover, the assessment of the persistence of allogenic CAR T without TCR is a major issue since the community of hematologist is anxious to know if these CARs without functional TCR will be able to persist in vivo and have long-term action, which is a crucial point of CAR therapies, now known.

The report is well written and experiments well controlled.

However a study about potential on tumor/off target toxicity should be done, even if UCART123 have a safety switch, mainly because it functionality has not been demonstrated in the report and because a patient died with this/or another version of UCART123. Moreover, testing of in vivo persistence would be of great interest.

Response: We appreciate Reviewer’s thoughtful comments and concerns. With respect to toxicity to normal cells, we have performed colony-forming assays and have demonstrated minimal toxicity of UCART123 against normal hematopoietic cells in vitro (**Extended data Fig. 6**).

Extended data Fig. 6

Extended data Fig. 6. The toxicity of UCART123 against normal hematopoietic cells *in vitro*. (a) Normal BM-derived hematopoietic cells were co-cultured with either UCART123 cells or non-transduced TCR $\alpha\beta$ -deficient T cells (TCR $\alpha\beta$ KO) for 16 hours and counted by flow cytometry. Each point represents the data obtained from triplicate experiments (using 3 independent donors), and the mean \pm SD value is shown. (b) Normal BM-derived hematopoietic stem cells were co-cultured for 2 weeks with UCART123 at different ratios and the cultures analyzed by colony-formation assay. Erythroid colony-forming units (CFUs) and myeloid CFUs were counted separately.

Please further refer to the co-submitted manuscript by Mayumi Sugita, which presents the effect of UCART123 on normal hematopoietic cells *in vivo* using humanized xenografts with both CD34⁺ CB cells (N=2) and lymphocyte-depleted BM cells from healthy donors. This data (**Extended data Fig. 4e**, see below) indicate that UCART123 therapy does not significantly impact normal CD34⁺ progenitor cells.

Extended data Fig. 4e from Mayumi Sugita's paper

Extended Data figure 4. Evaluation of subsets after UCART123 treatment in humanized mouse with normal hematopoiesis. Humanized mice engrafted with CD34⁺ CB cells (CB-I, CB-II) or normal bone marrow (nBM) were treated with 10×10^6 or 2.5×10^6 UCART123 cells or TCR $\alpha\beta$ KO T cells (10 million of each for CBs cohorts and 2.5 million of each for nBM cohort). After 1-3 months post treatment, mice were sacrificed and remaining subsets in BM were evaluated by flow cytometry. **e**, changes in the proportions of hematopoietic progenitor cells (CD34⁺) were evaluated. Each symbol represents an animal in each of the experiments and cohorts. Bar represents

the mean with the SD of all cohorts. one-way ANOVA.

With respect to toxicity against endothelial and other normal cells, this was not tested in our study. However, preliminary results from autologous CD123 CART clinical trial at the City of Hope¹ indicated lack of bone marrow or other organ toxicity, and no evidence for capillary leak syndrome. Similarly, in an ongoing clinical trial with the antibody-drug conjugate against CD123 IMGN632, no endothelial or marrow toxicity was seen in AML or BPDCN patients (Discussion, Page 11)^{2,3}.

The question of persistence *in vivo* of CARTs lacking functional TCR is extremely important. To address this in our pre-clinical study, we have examined the persistence of UCART123 cells in the blood, spleen, and BM of mouse injected with 10×10^6 UCART123 and sacrificed on day 78 (57 days after UCART123 cell injection). UCART123 cells could be readily detected in the spleen (16.4% CAR+ cells among viable cells; **Extended Data Fig. 8**) and in the bone marrow (1.1% CAR+ cells; **Extended Data Fig. 8**) of mouse injected with PDX-1, indicating long-term persistence. Additionally, we performed digital droplet PCR (ddPCR) in spleen cells from PDX-3 injected with UCART123. UCART123 cells were detectable in the mice that died 225 days after CART cells injection, supporting long-term persistence (**Extended Data Fig. 11**).

We have further added in Discussion the following:

It should be noted that the persistence in humans is expected to be limited by the rejection of the allogenic product upon recovery from lymphodepletion, which cannot be tested in the immunodeficient mouse models. Encouragingly, according to recent reports in relapsed/refractory B-ALL treated with UCART19⁴ and in NHL patients treated with ALLO-501⁵, both products bearing *TRAC* knock-out generated by TALEN technology, showed high response rates and allogenic CAR-T cell expansion, supporting the notion that even limited CAR-T cell *in vivo* persistence is sufficient to produce meaningful responses.

Extended data Fig. 8

Extended data Fig. 8. The presence of UCART123 cells was analyzed in the blood, spleen, and bone marrow of the mouse treated with 10×10^6 UCART123 cells (PDX-1 model) and sacrificed on Day 78 (57 days after UCART123 treatment). UCART123 cells were detected by flow cytometry using CD123-Fc protein and an anti-mouse Fc antibody.

Extended data Fig. 11: Detection of UCART cells in spleen of PDX-3 model mice by ddPCR.

Treatment	CD123 expression	Days after CART injection	UCART count	ABL1 count	UCART/ABL1*10000
RNA1 Untreated	CD123+	23	3	72,642	0.4
RNA2 Untreated	CD123+	23	1	57,438	0.2
RNA3 UCART123 10M group	CD123-	188	1	70,236	0.1
RNA4 UCART123 1M group relapse	CD123-	225	13	68,332	1.9
RNA5 UCART123 10M group	CD123-	225	575	67,144	85.6
RNA10 UCART123 1M group	CD123+	119	857	42,772	200.4
RNA13 UCART123 1M group	CD123-	100	37	9,776	37.8
RNA16 UCART123 1M group relapse	CD123-	71	43	21,939	19.6
RNA17 UCART123 1M group relapse	CD123-	78	4	1,510 *	26.5
NTC * NA	NA	NA	1	43 *	232.6

* NTC: no template control.

* Low ABL1 count less than 5,000.

UCART123 transcripts in spleen were evaluated with digital droplet PCR (ddPCR) at the time point of euthanasia due to sickness or disease progression. Copy numbers of UCART123 transcript relative to 1.0x 10⁴ ABL1 transcripts (reference) were calculated.

The lentiviral construct contains the safety switch RQR8. The functionality of this system is not discussed in vitro nor in vivo.

Response: The ability to eliminate RQR8+ T-cells through Complement-dependent Cytotoxicity (CDC) following treatment with rituximab (RTX), a therapeutic monoclonal antibody specific for CD20, was examined. As shown in **Extended data Fig. 1a**, RQR8+ T-cells are specifically eliminated in the presence of both RTX and complement.

We further studied the ability of RTX to deplete RQR8+ cells through Antibody Dependent Cell-mediated Cytotoxicity (ADCC), by co-culturing of UCART123 cells with NK cells in the presence or absence of RTX. Results presented in Figure **Extended data Fig. 1b** demonstrate reduction of the viable UCART123 cells at 1:1 E:T ratio. It should be noted that in a study by Pule et al.⁶ RQR8-expressing T-cells were efficiently depleted *in vivo* by rituximab reengineered to mouse IgG2a in blood, spleen, bone marrow, and lymph nodes.

Extended data Fig. 1

Extended data Fig. 1. Rituximab mediated depletion of UCART123. Panel a shows the relative percentage of CAR+ cells detected after 2h incubation of UCART123 with Rituximab (RTX), baby rabbit complement (BRC) or both, indicating efficient depletion through RQR8 by CDC. Panel b shows the efficacy of ADCC when UCART123 cells were co-cultured for 24h with NK cells in the presence or not of RTX at different NK:UCART123 ratios. In both cases, the data presented was obtained using large scale batches of UCART123, in four independent experiments.

Please further refer to the co-submitted manuscript by *Mayumi Sugita*, showing recurrence of the leukemia following that rituximab administration post CART therapy, supporting the ability of rituximab to eliminate UCART123 cells *in vivo* (**Figure (Reviewer)#1**).

Figure (Reviewer)#1 from Mayumi Sugita's Response to Reviewers

Figure (Reviewer)#1. UCART123 cell can be eliminated by rituximab. MOLM13-BLIV bearing mice (NSG) were treated with Saline (N=3) or 3M UCART123 cells (N=14). Seven days after treatment the 14 mice were randomized into two groups, Saline (N=6) and Rituximab 10mg/kg (N=8) and were treated days 7-11 and days 37-41. Leukemia was monitored by radiance. Figure shows a representative example of the radiance measured at day 52 in the surviving UCART123 treated animals showing disease progression in the group treated with rituximab.

Figure 2a-b: All of the 8 BPDCN patients express similar and high level of CD123, however UCART123 demonstrated cytotoxicity against 5 of the 8 samples, what are hypothesis of the authors about this lack of toxicity on CD123+ blasts? It may be interesting to put CD123 MFI of CAL-1 cells on the graph 2a to compare with BPDCN patients.

Response: The Reviewer is correct, we were unable to demonstrate significant correlation between CD123 MFI and specific cell lysis of 8 BPDCN patients' samples, see **Figure (Reviewer)#2** below. It is possible that a short-term 24-hr in vitro cytotoxicity assay is insufficient to evaluate a full impact of CART therapy. In addition, the level of cytotoxicity observed may be influenced by the presence of different levels of stimulatory or inhibitory cell surface proteins. It should be noted that we generated 3 PDX models from 3 of the 8 BPDCN patients. The specific lysis of these 3 samples *in vitro* was 77.9%, 22.4% and 73.9%, respectively. However, despite differences in potency *in vitro*, UCART123 eliminated CD123+ BPDCN cells *in vivo* in all three BPDCN PDX (**Fig. 3, 4 and 5**). The CD123 MFI of CAL-1 cells is included in the revised **Figure 2a** and is in the same range as CD123 expression in primary BPDCN samples.

Figure (Reviewer)#2

Figure (Reviewer)#2. No significant correlation between CD123 MFI and specific cell lysis of 8 BPDCN patients' samples.

Figure 1b and 2b : The authors presented specific lysis of UCART123, I think it could be interesting to present cell death after co-culture with TCRαβ KO T cells in comparison with co-culture UCART123 allowing to show potential alloreactivity as it is done for degranulation experiments and IFN-γ secretion.

Response: In the cytotoxicity assays presented in Figure 2B, the percentage of specific cell lysis was calculated by comparing the reduction in viability of target cells when co-cultured with UCART123 to that observed following co-culture with TCRαβ KO T cells. The results show high level of cytotoxicity of UCART123 cells when compared with TCRαβ KO T cells in 5 of the 8 primary BPDCN samples tested. We do not have the data for the viability of the target cells incubated alone. However, as we do not see degranulation or interferon gamma secretion with the TCRαβ KO T cells co-cultivated with these same target cells support absence of alloreactivity. This data are now included in a new **Extended data Fig. 5**.

Extended data Fig. 5

Extended data Fig. 5. Antitumor activity of UCART123 against primary BPDCN samples in vitro. Viability of MOLM13 cells and of the primary BPDCN samples upon co-culture for 16 hours with either UCART123 cells or non-transduced TCRαβ-deficient (TCRαβ KO) T cells. Each point represents the data obtained from triplicate experiments, and the mean ± SD values are shown. *P≤0.05, **P≤0.01, ***P≤0.001.

Figure 3: The level of BPDCN engraftment in blood before CAR injection might be detailed to

control the level of the different groups. Tumor burden might be given in absolute values (blasts/microL) to compare precisely evolution before and after CAR injection.

Response: Circulating tumor burden was measured by flow cytometry in peripheral mouse blood samples collected on the indicated days. We agree that demonstrating absolute values of blasts could be more informative. Unfortunately, these data were not collected.

Extended data figure 3A: mouse of the 10×10^6 UCART123 treatment group died, the authors did not detect BPDCN cells in the BM so they conclude that the death is not due to BPDCN progression. Do they study other organs such as spleen or lungs?

Response: We performed necropsy and flow cytometric analysis of tissues from bone marrow and spleen and failed to detect BPDCN cells (**Extended data Fig. 7**). We did not evaluate lungs but macroscopically they looked normal.

Extended data Fig. 7

Extended data Fig. 7. no BPDCN cells (hCD123+ and hCD56+ cells) were detected in bone marrow or spleen of died mouse from 10×10^6 UCART123 treatment group. Cells were isolated from bone marrow or spleen of UCART123-treated mouse when it died. Expression of hCD123+ and hCD56+ was measured by flow cytometry.

Figure 4, PDX models: it is not explained why do authors used sometimes NSG mice and sometimes NSG-S mice especially for BPDCN-3 cells. Authors observed CRS syndrome in NSG mice and not in NSG-S mice, the reason of that observation might be suggested. Concerning the CRS observed, as the construct have the RQR8 safety switch? if yes, the use of rituximab to diminish the CRS might be tested

Response: Following initial reports of better permissive environment of humanized NSG-S mice, we compared NSG and NSG-S mice in preliminary PDX-generating experiments and did not find significant differences in the speed of engraftment between these strains. These mice were subsequently used interchangeably as recipients in efficacy studies depending on mice' availability (but same strain was always used for selected study). If anything, one would expect higher CRS in NSG-S mice expressing human cytokines, but we observed CRS in NSG mice.

The Reviewer is correct in that rituximab could be used to eliminate the UCART123 harboring RQR8 safety switch, but this was not tested in this model. As mentioned above, in a study by Pule et al.⁶ RQR8-expressing T-cells were efficiently depleted *in vivo* by rituximab reengineered to mouse IgG2a, in blood, spleen, bone marrow, and lymph nodes. In a co-submitted manuscript by Sugita et al., the leukemia was recurrent in animals treated with UCART123, indicating the capacity of rituximab to eliminate UCART123 cells *in vivo* (see **Reviewer Figure #1**, Page 4).

Reviewer #3 (Remarks to the Author):

The authors characterize the performance of their original healthy donor-derived CAR T cells targeting CD123 on BPDCN, with *in vitro* killing assay and PDX models. The CAR T cells were constructed with the TRAC constant gene disrupted by the TALEN technology. Multiple issues need to be overcome for the success of allogeneic CAR T cell therapy directing CD123 on AML and BPDCN; 1) anti-tumor cell efficacy, 2) control of GVHD, 3) measures against allogeneic immune-related eradication of CAR T cells, and 4) intentional eradication of CAR T cells to protect normal hematopoietic cells. The authors' CAR T cells are constructed aiming at above 1), 2), and 4). In the current manuscript, however, only the anti-tumor cell efficacy is evaluated, although TCR gene disruption to eventually aiming at avoiding GVHD may have helped evaluating the anti-tumor activity in the PDX models.

Major

1. There is an obvious limitation in the manuscript as indicated above. The authors should clearly define that their purpose is to show the aspect of anti-tumor activity of their CD123 CAR T cells, discriminating from other issues. Then, they need to describe comparison between the preceding anti-CD123 CAR T cells and theirs simply in terms of the efficacy. To this end, it is important to describe the anti-tumor activity and the expression levels of CD123. The *in vitro* data do not show the correlation between the efficacy and the expression levels of CD123 among BPDCN samples.

Response: We had previously performed *in vitro* experiments to evaluate activity of UCART123 against AML cell lines with different CD123 expression levels, by measuring degranulation, cytotoxic activity and IFN γ release. The results shown in **Extended data Fig. 3** demonstrate lack of activity of CART on CD123-negative Daudi cells; and a trend for increasing responses correlating with the levels of CD123 antigen expression by the target cells.

Extended data Fig. 3: In vitro activity of T-cells expressing CAR123 against cell lines expressing different levels of CD123. The upper row of **panel A** shows the number of CD123 molecules at the cell surface, using the Qifikit surface antigen quantification kit. Daudi cells were used as a CD123 negative control, while KG1a, MOLM13 and RPMI-8226 were used as targets expressing increasing levels of CD123. The table on the upper right portion shows the mean and the SD values from 6 independent staining. Representative flow cytometry data is shown in the lower row of **panel A**, with blue histograms corresponding to cells labelled with anti-CD123 monoclonal antibody (clone 6H6), while the grey histograms correspond to cells labelled with the isotype control. **Panel B** shows degranulation activity of CAR123 cells, either cultured alone or co-cultured during 6h with the different cell lines described in panel A, at a 1:1 E:T ratio. Degranulation was evaluated by flow cytometry after staining with CD8 and CD107a antibodies. The degranulation of the CD8⁺ and CD8⁻ fractions (considered as CD4⁺ cells) is shown. **Panel C** shows IFN γ levels released in the supernatants of overnight co-cultures of CAR123 cells with target cells at 1:1 E:T ratio, using an ELISA test. Cytotoxic activity against MOLM13 and RPMI-8226 cells is shown in **panel D**. Co-cultures were settled with each of the CD123⁺ cell lines (together with Daudi CD123^{neg} cells as an internal negative control) at a 10:1 effector to target (E:T) ratio for 18 hours. Cytotoxic activity was evaluated by assessing the viability of the different target cell populations by flow cytometry at the end of the co-culture. Cell killing activity was normalized to the activity against CD123⁻ Daudi cells.

NTD: non transduced T-cells. **UCART123:** T-cells transduced with CAR123, transfected with TRAC targeting TALEN[®] and purified by depletion of remaining TCR $\alpha\beta$ ⁺ T-cells. **CART123:** T-cells transduced with CAR123, mock transfected (TCR $\alpha\beta$ ⁺ cells). The experiments shown in panels B-D were performed with T-cells from three different donors, each of which is represented by a different color (black, green or white symbols). The donor highlighted in black was transduced with a R&D backbone, while the two other donors were transduced with an rLV produced from the same backbone used to manufacture UCART123 clinical batches. Nevertheless, all three donors express the CAR from the same EF1 α -RQR8-2A-CAR123 lentiviral expression cassette.

2. The authors should describe whether they found a significant effect of TCR depletion in the PDX experiment. Because the PDX models are important for the evaluation of many CAR T cells, it may be helpful if they compare UCART123 with the version holding TCR.

Response: This comparison was not performed in the PDX experiments as the proof of concept for the activity of UCART123 cells compared to that of TCR $\alpha\beta$ ⁺ cells expressing the same CD123CAR had been done previously using AML cell lines as targets. It had previously been shown that the TRAC KO, in the context of a CD19 CAR, has no impact on CAR T-cell activity⁷. The *in vitro* data is summarized in **Extended data Fig. 3**, in which the activity of CAR123 expressing T-cells, whether or not engineered to lack TCR $\alpha\beta$ expression at the cell surface, show comparable activity in all assays and for the three donors evaluated.

The objective of this manuscript was indeed to provide preclinical proof of concept for the allogeneic approach of CD123 targeted CAR T-cells and not to evaluate the impact of the TRAC gene KO. Besides the benefits related to an off-the-shelf therapy, an additional interest of UCART123 relies in the fact that CD123 is also expressed in normal hematopoietic progenitors, which raises a safety concern in the context of autologous anti-CD123 CAR T-cells. In fact, even if the data in the manuscript co-submitted by Mayumi Sugita/Dr. M. Guzman evaluating the effect of UCART123 on normal hematopoietic cells *in vivo* shows reduced toxicity of UCART123, we consider that the expression of the same CAR in autologous cells, which will persist longer than allogeneic cells, could increase the risk of prolonged myelosuppression.

In vivo experiments were also performed in which activity of UCART123 and CART123 cells was evaluated for their ability to suppress MOLM13 tumor progression in NOG mice. The results, shown for the reviewer below (**Figure (Reviewer)#3**), indicate comparable activity and increased survival compared to control animals, for both edited and non-edited CAR123 T-cells.

Figure (Reviewer)#3

Figure (Reviewer)#3: *In vivo* activity of T-cells expressing CAR123 in NOG mice harboring MOLM13 tumors. The capacity of CART123 and UCART123 cells to control tumor progression *in vivo* was evaluated by bioluminescence analysis on a weekly basis during 5 weeks. Experiment was terminated on day 60. 2.5×10^5 MOLM13 tumor cells were intravenously injected at D -7, CAR123 expressing cells were intravenously injected at D 0 (10 million per mouse).

3. It is important to clarify the frequency of the downregulation of CD123 by genetic events as the mechanism of the acquisition of resistance, particularly if there is a possibility of differences in the frequencies depending on the anti-CD123 treatment modalities including SL-401.

Response: We are not aware of the reported loss of CD123 expression in patients that relapse or are resistant to SL-401 (tagraxofusp). The recent report from Dr Andrew Lane, a co-author on our study, interrogated resistance to tagraxofusp and found that it was not associated with CD123 loss, in patients or in model systems.⁸ They further identified as a key resistance mechanism methylation of DPH1 enzyme, which impairs the tagraxofusp's action to ADP-ribosylate cellular targets. Further, our report at ASH 2020 (Pemmaraju et al³, Clinical Profile of IMGN632, a Novel CD123-Targeting Antibody-Drug Conjugate (ADC), in Patients with Relapsed/Refractory (R/R) BPDCN) using anti-CD123 ADC IMGN632 demonstrated responses in 29% of patients with relapsed BPDCN progressing post tagraxofusp, all of whom harbored CD123. Hence, loss of CD123 described here in one PDX model, could be related to the genetic characteristics of the BPDCN sample where this was observed, which could have favored the emergence of mutant forms following the selective pressure of UCART123 treatment.

Minor

1. The major evaluation by PDX models were performed with 3 patients-derived samples. Whereas the authors described PDX model 1, 2, and 3, they used samples from patients #1-3 for No1, #3 for No2, and #2 for No3 experiment with the completely the same or very similar experimental designs to evaluate the reproducibility. It should not be designated as a “model” but just “experiment.”

Response: Thank you. We corrected it.

Reviewer #4 (Remarks to the Author):

In the present manuscript the authors present preclinical data for an off-the shelf CAR-T product directed against CD123. Overall, the manuscript is very well written and the data presented in a clear way. However, preclinical data for CAR-T targeting CD123 for treatment of AML or BPDCN have been previously presented by other groups. Hence, the manuscript lacks novelty in regard of the therapeutic approach. The emergence of a CD123 escape mutant in the in vivo model is interesting for the clinical translation of the approach and should be further investigated. Is the emergence of escape mutants specific to the applied CAR-T product or maybe related to the used CD123 binder used for the CAR construct?

Response: While we agree that CD123 is not a novel target for AML and BPDCN, the co-submitted papers by our group and by Mayumi Sugita/Dr. M. Guzman represents a first pre-clinical report of the activity of the “universal” allo-CART in pre-clinical models of AML and BPDCN. With respect to CD123 escape mutants, we believe it could be related to the anti-CD123 targeted therapy, similar to reported silencing of CD19 or CD22 targets in B-ALL with CART approaches. The most plausible explanation of CD123 escape is the selection of a small pre-existing subclone(s) lacking CD123. The diversity of the genomic lesions found indicate possible additional mechanisms by which anti-CD123 UCART123 was associated with CD123 gene loss.

The CD123 targeting moiety of the CAR is derived from a monoclonal antibody different from the ones used in preclinical works published by other groups. This specific binder was selected in order to minimize on-target/off-tumor toxicities against normal hematopoietic progenitors (further described in the co-submitted paper by Mayumi Sugita/Dr. M. Guzman). Even if this could also relate to the emergence of escape mutants, the authors consider that this observation might most probably be related to the genetic characteristics of the BPDCN sample where this was observed, which could have favored the emergence of mutant forms following the selective pressure of UCART123 treatment.

The translational value of the presented data remains unclear, as UCART123 is/was already under clinical investigation for BPDCN. A study hold was put on a study after death of the first patient treated with UCART123. Public information on the current status of the study is lacking. In the present manuscript it is described that UCART123 harbors as safety switch (Rituximab target epitope). Is the safety switch non-function or why did it not work in the clinic situation?

Response: In response to this comment and also Reviewer #2 (see page 3), we have now included *in vitro* data indicating that rituximab eliminates RQR8+ T-cells through CDC and ADCC mechanisms (**Extended data Fig. 1**). In a co-submitted manuscript by Sugita et al., the leukemia was recurrent in animals treated with UCART123, indicating the capacity of rituximab to eliminate UCART123 cells *in vivo* (see **Reviewer Figure #1**, Page 4). So far, Rituximab has not been yet utilized in AML or BPDCN patients treated on UCART123 clinical trial (Roboz ASH 2020)⁹.

The manuscripts lack any data how to tackle such acute toxicities and also lack any data/explanation how long-term toxicities against CD123+ expressing hematopoietic progenitor cells should be mitigated. Adding such data would enhance the scientific value of the manuscript as in the present manuscript the pathway towards further clinical translation of the approach is unclear.

Response: Please kindly refer to the Response to Reviewer #2 page 1, where we summarize the results of the *in vitro* and reference *in vivo* data in a co-submitted manuscript by Mayumi Sugita et al., demonstrating lack of hematological toxicity.

It should be noted that while in ongoing early stage clinical trials with autologous CART and anti-CD123 monoclonal antibodies the myelosuppression has not been observed, FDA mandates availability of the potential allogenic stem cell transplant donor for any patient undergoing anti-CD123 CART therapy.

In Figure 1 and Figure 2, CD107a up-regulation is shown for CD8+ UCART123 cells, but data is lacking for CD4+. Are CD4+ cells contributing to target cell lysis in UCART123 products as shown by other groups for their CD123+ CAR-T? What is the CD4/CD8 composition of the tested UCART123 batches?

Response: We appreciate the Reviewer’s point that in recent papers there is evidence of the role for CD4+ CART in anti-tumor efficacy. In our study, these assays were performed before these papers were published, and we focused on CD8+ CART cells, which were well established to exhibit potent anti-leukemia toxicity. The data on CD4+ CART responses was not initially collected. Nevertheless, we have reanalyzed the degranulation data and, by gating on the CD8^{neg} fraction of T-cells, we confirm the degranulation activity against CD123 target cells. This data has been added in **Extended data Fig. 3**, panel B.

Extended data Fig. 3. In vitro activity of T-cells expressing CAR123 against cell lines expressing different levels of CD123. Panel B shows degranulation activity of CAR123 cells, either cultured alone or co-cultured during 6h with the different cell lines described in panel A, at a 1:1 E:T ratio. Degranulation was evaluated by flow cytometry after staining with CD8 and CD107a antibodies. The degranulation of the CD8+ and CD8- fractions (considered as CD4+ cells) is shown.

For the UCART123 batches tested in the present work, CD4/CD8 ratio were ranged from 0.6 to 1.2 in large scale batches, and from 0.3 to 3.3 in small scale batches, see **Table** below).

Reviewer Table 1: CD4/CD8 ratios in UCART12 batches

Large scale batches:

	Pr62 (UCART123)	Pr66 (NTD)	Pr67 (UCART123)	Pr67 (NTD)
% CAR+	90,5	-	93,7	-
% CD4	36,0	38,4	54,2	37,7
% CD8	58,7	57,7	44,8	32,7
CD4/CD8 ratio	0,61	0,67	1,21	1,15

Small scale batches:

	d1303		d1704		d2205	
	CART123	UCART123	CART123	UCART123	CART123	UCART123
% CAR+	98,7	97,5	99,2	99,0	97,2	96,7
% CD4	43,7	47,3	75,6	76,0	25,9	19,2
% CD8	54,0	49,1	23,2	22,9	67,6	66,6
CD4/CD8 ratio	0,81	0,96	3,25	3,32	0,38	0,29

In the second in vivo model experimental mice treated with UCART123 died early likely due to CRS. Repeating the same experiment with a lower number of leukemic cells and earlier on-set of treatment resulted in long-term survival of treated mice. However, using the same number of leukemic cells and starting treatment at a similar time point as in the experiment shown in Fig. 4A did not results in CRS. The reviewer concludes that sever CRS is obviously very much dependent on the leukemia kinetic in vivo and maybe specific interaction between the UCART123 and the leukemic cells. However, this problem will be very likely also encounter in the clinical setting. The authors should at least provide an explanation in the discussion section how they envisage to encounter acute toxicities as already observed with UCART123 in the clinic.

Response: We fully agree with the Reviewer that CRS remains the expected toxicity with this and other CART constructs. We have reported at ASH 2020 that the 1st dosing cohort of AMELI-01 trial of UCART123 in patients with relapsed or refractory AML has cleared safety without dose-limiting toxicity, and enrollment at the next dose levels is ongoing. In this study, management of CRS is per institutional guidelines, and it envisions use of rituximab to activate a built-in suicide mechanism (RQR8).

References

1. Budde LE, *et al.* Abstract PR14: CD123CAR displays clinical activity in relapsed/refractory (r/r) acute myeloid leukemia (AML) and blastic plasmacytoid dendritic cell neoplasm (BPDCN): Safety and efficacy results from a phase 1 study. *Cancer Immunology Research* **8**, PR14-PR14 (2020).
2. Daver NG, *et al.* Clinical Profile of IMG632, a Novel CD123-Targeting Antibody-Drug Conjugate (ADC), in Patients with Relapsed/Refractory (R/R) Acute Myeloid Leukemia (AML) or Blastic Plasmacytoid Dendritic Cell Neoplasm (BPDCN). *Blood* **134**, 734-734 (2019).
3. Pemmaraju N, *et al.* Clinical Profile of IMG632, a Novel CD123-Targeting Antibody-Drug Conjugate (ADC), in Patients with Relapsed/Refractory (R/R) Blastic Plasmacytoid Dendritic Cell Neoplasm (BPDCN). *Blood* **136**, 11-13 (2020).
4. Benjamin R, *et al.* Genome-edited, donor-derived allogeneic anti-CD19 chimeric antigen receptor T cells in paediatric and adult B-cell acute lymphoblastic leukaemia: results of two phase 1 studies. *Lancet* **396**, 1885-1894 (2020).
5. Neelapu SS, *et al.* First-in-human data of ALLO-501 and ALLO-647 in relapsed/refractory large B-cell or follicular lymphoma (R/R LBCL/FL): ALPHA study. *Journal of Clinical Oncology* **38**, 8002-8002 (2020).
6. Philip B, *et al.* A highly compact epitope-based marker/suicide gene for easier and safer T-cell therapy. *Blood* **124**, 1277-1287 (2014).
7. Poirot L, *et al.* Multiplex Genome-Edited T-cell Manufacturing Platform for "Off-the-Shelf" Adoptive T-cell Immunotherapies. *Cancer Res* **75**, 3853-3864 (2015).
8. Togami K, *et al.* DNA methyltransferase inhibition overcomes diphthamide pathway deficiencies underlying CD123-targeted treatment resistance. *J Clin Invest* **129**, 5005-5019 (2019).
9. Roboz GJ, *et al.* Ameli-01: Phase I, Open Label Dose-Escalation and Dose-Expansion Study to Evaluate the Safety, Expansion, Persistence and Clinical Activity of UCART123 (allogeneic engineered T-cells expressing anti-CD123 chimeric antigen receptor), Administered in Patients with Relapsed/Refractory Acute Myeloid Leukemia. *Blood* **136**, 41-42 (2020).

Reviewers' Comments:

Reviewer #2:

Remarks to the Author:

The authors generated allogeneic T cells expressing a second generation anti-CD123 chimeric antigen receptor and tested them against a rare leukemia called BPDCN that expresses high levels of CD123.

Authors make revision that partially respond to my comments.

Comments:

The novelty of this study is the use of an allogenic CAR T cells with TALEN nuclease edition and in providing a mechanistic study about the loss of CD123 observed in one of three BPDCN PDX studied.

However, no toxicity study has been reported here whereas it seems to be mandatory in the context of allogenic cells and since CD123 is expressed on some hematopoietic stem cells, endothelial cells, monocytes and basophils.

Authors show no cytotoxicity of UCAR123 on CD34+ cells in vivo using colony forming assay (no statistics are presented) and humanized xenografts (co-submitted manuscript by Mayumi Sugita). There is no data on monocytes and endothelial cells, so the answer is partial.

The lentiviral construct contain the safety switch RQR8. The functionality of this system is not discuss in vitro or in vivo.

Authors demonstrated in vitro functionality, but in vivo demonstration lacks conviction. In fact, they showed recurrence of the leukemia following rituximab administration post CART therapy, but then did not demonstrated elimination of CAR-T cells. It is necessary to evaluate if the RQR8 treatment lead to rapid CAR elimination in vivo (or at least cytokines decrease) : it is what we need to treat serious adverse events.

Figure 2a-b : All of the 8 BPDCN patients express similar and high level of CD123, however UCART123 demonstrated cytotoxicity against 5 of the 8 samples, what are hypothesis of the authors about this lack of toxicity on CD123+ blasts ? It may be interesting to put CD123 MFI of CAL-1 cells on the graph 2a to compare with BPDCN patients.

It is ok, MFI of CAL-1 has been added. No correlation is observed between CD123 MFI and cytotoxicity, so there is no explanation for low cytotoxicity of UCAR123 against some BPDCN cells. Absence of cytotoxicity observed against some sample is explain by a possible short-term 24-hr in vitro cytotoxicity assay. I don't agree with this hypothesis. In fact, at a 10/1 ET ratio during 24h, cytotoxicity should be higher, it is a very favorable condition.

Authors also hypothesis that the low cytotoxicity level may be influenced by the presence of some stimulatory or inhibitory cell surface proteins. This hypothesis should be confirm by flow cytometry staining for example.

Figure 1b and 2b : The authors presented specific lysis of UCART123, I think it could be interesting to present cell death after co-culture with TCR $\alpha\beta$ KO T cells in comparison with co-culture UCART123 allowing to show potential alloreactivity as it is done for degranulation experiments and IFN- γ secretion.

Cell death after co-culture with TCR $\alpha\beta$ KO T cells in comparison with co-culture UCART123 has been added on graph: TCR $\alpha\beta$ KO T cells demonstrate an important cell death with no explanation. Statistical analysis showed significant results however cell death difference between co-culture with TCR $\alpha\beta$ KO T cells in comparison with co-culture UCART123 is very low for 7 PDX of 8 (<10% of more cytotoxicity with UCAR123). This observation raises questions about the effectiveness of the UCART123 (extended data fig 5).

Figure 3 and extended data figure 3: A mouse of the 10 \times 10⁶ UCART123 treatment group, the authors did not detected BPDCN cells in the BM so they conclude that the death is not due to BPDCN progression. Do they study other organs such as spleen or lungs ?

It is ok, thank you for these data.

Figure 4, PDX models : why do authors used sometimes NSG mice and sometimes NSG-S mice especially for BPDCN-3 cells. Authors observed CRS syndrome in NSG mice and not in NSG-S mice. Concerning CRS observed, as the construct have the RQR8 safety switch, is it possible to use rituximab to diminish the CRS?

I think that use of rituximab to demonstrate a decrease in CRS would have been a major element

(using humanized mouse model for example). Authors cannot demonstrate this comment. Authors have only argued this point, no experiment have been done. In vivo demonstration of UCART123 elimination after Rituximab treatment is very important for clinical development and this point is not fully addressed in this work.

Reviewer #3:

Remarks to the Author:

The revised manuscript is much improved through the authors' efforts to respond to the reviewers' criticisms. Now readers can much more easily understand the significance of the manuscript. At the same time, however, the readers can also perceive the limitations of this paper in terms of the novelty, particularly seeing the recent progresses in the filed, including the clinical trials of UCART and similarly generated allogeneic CART cells.

Minor

The references for RQR8 should be appropriately cited; Blood 2014;124:1277 may be the right one.

Reviewer #4:

Remarks to the Author:

The reviewer grateful acknowledges the answer and additional data provided by the authors, which clearly improved the manuscript. However, the reviewer still misses any reference to the previous clinical experience with UCART123 and its relationship to the presented experimental data. Is it the same product? Are their differences? What clinical insights in terms of safety and efficacy were obtained so far? It would clearly strengthen the novelty and significance of the manuscript if the authors address this point.

Minor point:

Extended data set Fig. 3: y-axis labeling should be corrected to CD8-CD107+ instead CD4+, as this describes the gating strategy correctly

REVIEWER COMMENTS

Reviewer #2 (Remarks to the Author):

The authors generated allogeneic T cells expressing a second generation anti-CD123 chimeric antigen receptor and tested them against a rare leukemia called BPDCN that expresses high levels of CD123.

Authors make revision that partially respond to my comments.

Comments:

The novelty of this study is the use of an allogeneic CAR T cells with TALEN nuclease edition and in providing a mechanistic study about the loss of CD123 observed in one of three BPDCN PDX studied.

However, no toxicity study has been reported here whereas it seems to be mandatory in the context of allogeneic cells and since CD123 is expressed on some hematopoietic stem cells, endothelial cells, monocytes and basophils.

Authors show no cytotoxicity of UCAR123 on CD34+ cells in vivo using colony forming assay (no statistics are presented) and humanized xenografts (co-submitted manuscript by Mayumi Sugita). There is no data on monocytes and endothelial cells, so the answer is partial.

Response: We appreciate Reviewer's thoughtful comments and concerns.

For the colony forming assay for normal CD34+ cells, we performed statistical analysis and there is no significant difference between UCART123 group and non-transduced TCR $\alpha\beta$ -deficient T cells group.

We were unable to generate data on healthy monocytes and endothelial cells. However, data reported in the co-submitted manuscript by Mayumi Sugita demonstrate that UCART123 did not cause severe myeloablation (**Extended Data fig. 4c**). In addition, evaluation of CD34+ progenitor cells after exposure to UCART123 indicate that hematopoietic progenitor cells were not ablated by UCART123 in the humanized mouse models (**Extended Data figure 4e**).

Extended data Fig. 4c and 4e from Mayumi Sugita's paper

Extended Data figure 4. Evaluation of subsets after UCART123 treatment in humanized mouse with normal hematopoiesis. Humanized mice engrafted with CD34+ CB cells (CB-I, CB-II) or normal bone marrow (nBM) were treated with 10×10^6 or 2.5×10^6 UCART123 cells or TCR $\alpha\beta$ KO T cells (10 million of each for CBs cohorts and 2.5 million of each for nBM cohort). After 1-3 months post treatment, mice were sacrificed and remaining subsets in BM were evaluated by flow cytometry. **c and e**, changes in the proportions of myeloid (CD33+, c) and hematopoietic progenitor cells (CD34+, e) were evaluated. Each symbol represents an animal in each of the experiments and cohorts. Bar represents the mean with the SD of all cohorts. one-way ANOVA.

The lentiviral construct contain the safety switch RQR8. The functionality of this system is not discussed in vitro or in vivo.

Authors demonstrated *in vitro* functionality, but *in vivo* demonstration lacks conviction. In fact, they showed recurrence of the leukemia following rituximab administration post CART therapy, but then did not demonstrate elimination of CAR-T cells. It is necessary to evaluate if the RQR8 treatment lead to rapid CAR elimination *in vivo* (or at least cytokines decrease): it is what we need to treat serious adverse events.

Response: To respond to this request by the Reviewer, we have conducted the *in vivo* study to evaluate if the RQR8 targeting leads to CAR-T depletion. The data included in Extended Data Fig. 10b demonstrate significant depletion of CAR-T cells in the murine spleens and bone marrow following rituximab administration post CAR-T therapy, supporting the functionality of the suicide switch *in vivo*.

Further, a profound reduction of IFN γ level was detected in peripheral blood samples from the mice treated with rituximab post CAR-T therapy (Extended Data Fig. 10c). These results further suggest that RQR8 targeting by rituximab leads to CAR-T elimination associated with abrogation of cytokine production.

Manuscript text was edited as following (page 7, Lane 14):

“To further evaluate if the RQR8 targeting could lead to rapid CAR-T elimination *in vivo* and dampen the cytokine release syndrome, we administered Rituximab 10mg/kg i.p. 2 days after UCART injections (day 19) for a total of 5 days (Extended Data Fig. 10a). Rituximab administration post CAR-T therapy caused significant depletion of CAR-T cells in the murine spleens and bone marrow (Extended Data Fig. 10b), supporting the functionality of the suicide switch *in vivo*. IFN γ level was similarly reduced in peripheral blood from the mice treated with rituximab post CAR-T therapy (Extended Data Fig. 10c). These results confirm that RQR8 targeting by rituximab leads to CAR-T elimination *in vivo* associated with abrogation of cytokine production.”

And in Discussion, p.11, lane 6:

“Our *in vivo* data confirm that RQR8 targeting by rituximab leads to CAR-T elimination *in vivo* associated with abrogation of cytokine production (Extended data Fig.10).”

Extended data Fig. 10. Rituximab depletes UCART123 *in vivo*. **a**, Experimental design using BPDCN-3 PDX cells. When engraftment was confirmed on day 17 after tumor cell injection, mice were randomized into 6 treatment groups (n=9 mice/group) and received treatment as follows: vehicle; 10×10^6 TCR $\alpha\beta$ KO T cells; 3×10^6 UCART123 cells; 3×10^6 UCART123 cells followed by Rituximab; 10×10^6 UCART123 cells or 10×10^6 UCART123 cells followed by Rituximab. Rituximab at 10mg/kg was administered i.p. 2 days after UCART injections (day 19) for a total of 5 days. **b**, Fractions of UCART123 cells in the spleen and bone marrow of mice

(n=3) from experimental cohorts. Mice were sacrificed on day 24 after tumor cell injection. UCART123 cells were detected by flow cytometry using CD123-Fc protein and an anti-mouse Fc antibody conjugated with PE. Compared to 10×10^6 UCART123 cells group: $*P \leq 0.05$. **c**, IFN- γ levels in peripheral blood of mice measured on day 19 and day 23. Compared to 10×10^6 UCART123 cells group: $****P \leq 0.0001$.

Additionally, in a co-submitted manuscript by Sugita et al., a separate study was performed to evaluate evaluation rituximab as a safety switch was performed in AML model. In this model, AML engrafted mice were treated with PBS, UCART123 cells or UCART123 cells followed by rituximab treatment starting on day 7 post leukemia injection. Data shown in **Fig. 1g,h** (see below) demonstrated that treatment with 10mg/kg of rituximab starting at 7 days after animals received UCART123 cells resulted in disease progression evaluated by luminescence, and decrease in detectable CAR T cells by flow cytometry, in contrast with the UCART123 mice that only received vehicle control and remained disease-free. Upon re-challenging mice with leukemia cells, mice treated with rituximab had significantly lower overall survival and more rapid disease progression compared with UCART123-pretreated mice that received vehicle and no Rituximab (**Extended Data fig.1e-f** in co-submitted manuscript, not shown here), demonstrating that rituximab eliminated residual UCART123.

Fig. 1 (Sugita et al).

g, MOLM13-BLIV engrafted mice were treated with PBS, UCART123 cells, or UCART123 cells followed by rituximab (RTX). Representative BLI images at pre-treatment, day 28 and day 42 measured are shown for each group. **h, top**, Frequencies of UCART123 cells (%) in mononuclear cells in peripheral blood on day 23 post UCART123 treatment were measured with flow cytometry. Each symbol represents one mouse and bar represents the average with the SD. $****p < 0.0001$, one-way ANOVA. **bottom**, Average radiance measured with BLI on day 28 and day 42 are shown. Each symbol represents one mouse and bar represents the average with the SD. $***p < 0.001$, Mann-Whitney test.

Figure 2a-b: All of the 8 BPDCN patients express similar and high level of CD123, however UCART123 demonstrated cytotoxicity against 5 of the 8 samples, what are hypothesis of the authors about this lack of toxicity on CD123+ blasts? It may be interesting to put CD123 MFI of CAL-1 cells on the graph 2a to compare with BPDCN patients.

It is ok, MFI of CAL-1 has been added. No correlation is observed between CD123 MFI and cytotoxicity, so there is no explanation for low cytotoxicity of UCART123 against some BPDCN cells.

Absence of cytotoxicity observed against some sample is explain by a possible short-term 24-hr in

vitro cytotoxicity assay. I don't agree with this hypothesis. In fact, at a 10/1 ET ratio during 24h, cytotoxicity should be higher, it is a very favorable condition.

Authors also hypothesis that the low cytotoxicity level may be influenced by the presence of some stimulatory or inhibitory cell surface proteins. This hypothesis should be confirmed by flow cytometry staining for example.

Response: We concur with the Reviewer that we could not identify the plausible explanation for lack of responses in some BPDCN samples treated with UCART123 *in vitro*. Realizing the limitations of the *in vitro* testing, we complemented these by three *in vivo* BPDCN PDX studies (Fig. 3, 4 and 5), which all demonstrate activity of the UCART123 product.

Figure 1b and 2b : The authors presented specific lysis of UCART123, I think it could be interesting to present cell death after co-culture with TCR $\alpha\beta$ KO T cells in comparison with co-culture UCART123 allowing to show potential alloreactivity as it is done for degranulation experiments and IFN- γ secretion.

Cell death after co-culture with TCR $\alpha\beta$ KO T cells in comparison with co-culture UCART123 has been added on graph: TCR $\alpha\beta$ KO T cells demonstrate an important cell death with no explanation. Statistical analysis showed significant results however cell death difference between co-culture with TCR $\alpha\beta$ KO T cells in comparison with co-culture UCART123 is very low for 7 PDX of 8 (<10% of more cytotoxicity with UCART123). This observation raises questions about the effectiveness of the UCART123 (extended data fig 5).

Response: As described above, due to rarity of the disease, we had to use frozen samples from the bank to increase the number of samples tested; unfortunately, the viability of these samples was compromised after thawing, which may have affected the results. There is a significant difference between co-culture with UCART123 in comparison with co-culture with TCR $\alpha\beta$ KO T cells. As above, we believe that the *in vivo* studies offer convincing support that the UCART123 product is highly effective in all three BPDCN models tested (Fig. 3, 4 and 5).

Figure 3 and extended data figure 3: A mouse of the 10 \times 10⁶ UCART123 treatment group, the authors did not detect BPDCN cells in the BM so they conclude that the death is not due to BPDCN progression. Do they study other organs such as spleen or lungs ?

It is ok, thank you for these data.

Figure 4, PDX models : why do authors use sometimes NSG mice and sometimes NSG-S mice especially for BPDCN-3 cells. Authors observed CRS syndrome in NSG mice and not in NSG-S mice. Concerning CRS observed, as the construct has the RQR8 safety switch, is it possible to use rituximab to diminish the CRS?

I think that use of rituximab to demonstrate a decrease in CRS would have been a major element (using humanized mouse model for example). Authors cannot demonstrate this comment. Authors have only argued this point, no experiment has been done. *In vivo* demonstration of UCART123 elimination after Rituximab treatment is very important for clinical development and this point is not fully addressed in this work.

Response: We now include new data from the *in vivo* experiment using Rituximab. Please kindly refer to the response above.

Reviewer #3 (Remarks to the Author):

The revised manuscript is much improved through the authors' efforts to respond to the reviewers' criticisms. Now readers can much more easily understand the significance of the manuscript. At the same time, however, the readers can also perceive the limitations of this paper in terms of the novelty,

particularly seeing the recent progresses in the field, including the clinical trials of UCART and similarly generated allogeneic CART cells.

Minor

The references for RQR8 should be appropriately cited; Blood 2014;124:1277 may be the right one.

Response: Thank you. The references were added as #18.

Reviewer #4 (Remarks to the Author):

The reviewer gratefully acknowledges the answer and additional data provided by the authors, which clearly improved the manuscript. However, the reviewer still misses any reference to the previous clinical experience with UCART123 and its relationship to the presented experimental data. Is it the same product? Are there differences? What clinical insights in terms of safety and efficacy were obtained so far? It would clearly strengthen the novelty and significance of the manuscript if the authors address this point.

Response: The trial with UCART123 is ongoing in AML patients (NCT03190278). The results have not yet been presented. The abstract as “Trial in progress” was presented at ASH 2020:

Session: 616. Acute Myeloid Leukemia: Novel Therapy, excluding Transplantation:

Authors: Gail J. Roboz, Daniel J. DeAngelo, David A. Sallman, Monica L. Guzman, Pinkal Desai, Hagop M. Kantarjian, Marina Konopleva, Nelli Bejanyan, Hany Elmariah, Francisco Javier Esteva, Andrew Garton, Kate Backhouse, Roman Galetto, Carrie Brownstein, and Naveen Pemmaraju

Title: *Ameli-01: Phase I, Open Label Dose-Escalation and Dose-Expansion Study to Evaluate the Safety, Expansion, Persistence and Clinical Activity of UCART123 (allogeneic engineered T-cells expressing anti-CD123 chimeric antigen receptor), Administered in Patients with Relapsed/Refractory Acute Myeloid Leukemia*

Background: Acute myeloid leukemia (AML) is the most common form of acute leukemia in adults, with an incidence that increases with age, and a generally poor prognosis. The prognosis remains especially grim for those who are older, have secondary AML, or relapsed or refractory (R/R) disease, in which 5-year OS is 5-10%. Therefore, novel therapeutic approaches are needed. CD123(IL3R α) is a cell surface target that is expressed on normal, committed hematopoietic progenitor cells, and a variety of hematological neoplasms, including AML, myelodysplastic syndrome (MDS), and blastic plasmacytoid dendritic cell neoplasm (BPDCN). UCART123 is genetically modified, allogeneic (“off-the-shelf”), anti-CD123 CAR T cell product candidate in which the TCR alpha constant gene is disrupted to reduce the risk of GvHD, and the CD52 gene is disrupted to permit the use of alemtuzumab for selective and prolonged host lymphodepletion. Also, the CAR is co-expressed with a suicide mechanism (RQR8), which can be activated by using rituximab. In vitro data have demonstrated that UCART123 efficiently targets primary AML cells, with minimal effect on normal progenitors. Also, in PDX mouse models of AML, UCART123 cells can eliminate tumor cells in vivo, prevent relapse, and improve survival; in a competitive BM/AML PDX model, UCART123 cells demonstrated preferential targeting of AML blasts (Guzman; Blood 2016).

Methods: AMELI-01 is a phase 1, multi-center clinical trial of UCART123 that employs an mTPI design to evaluate the safety, tolerability and preliminary anti-leukemia activity of UCART123 in patients (pts) with R/R AML. Additional objectives include determination of the MTD; characterization of the expansion, trafficking and persistence of UCART123; assessment of cytokine, chemokine and CRP levels after UCART123 infusion; and assessment of immune cell depletion, reconstitution and immune response. Dose escalation will include up to 28 pts. The dose expansion portion follows a Simon 2-stage design and will enroll up to an additional 37 pts. Eligible pts must be \leq 65 years of age with R/R AML, adequate organ function, a confirmed donor for potential back-up stem cell transplantation, and no ongoing $>$ G1 toxicity from prior treatment. Pts with APL, prior gene or cellular therapy, $>$ 1 allogeneic SCT, or those with a clinically relevant CNS disorder (including CNS leukemia) are not eligible. Pts receive a lymphodepletion (LD) regimen of either fludarabine and cyclophosphamide (FC) or fludarabine, cyclophosphamide plus alemtuzumab (FCA) starting on Day -5, followed by an infusion of UCART123 at one of 5 dose levels on Day 0. Pts are evaluated for the presence of dose-limiting toxicities (DLT) during a 28-day observation period, which extends to 42 days in the setting of marrow aplasia and/or persistent clinically significant cytopenias without residual AML. DL1 has cleared safety without DLT, and enrollment at the next dose levels are proceeding.

ClinicalTrials.gov Identifier: NCT03190278

Minor point:

Extended data set Fig. 3: y-axis labeling should be corrected to CD8-CD107+ instead CD4+, as this describes the gating strategy correctly

Response: Thank you. We corrected it.

REVIEWERS' COMMENTS

Reviewer #5 (Remarks to the Author):

Cai and colleagues have submitted a manuscript evaluating allogeneic TCR ko anti-CD123 CAR T cells for the treatment of blastic plasmacytoid dendritic cell neoplasm, a rare disease with a mostly poor prognosis. The authors provide in vitro and in vivo evaluation against both cell lines and patient derived models. The manuscript is overall well written and easy to follow. This manuscript has already undergone two rounds of revisions. Most of the data is sound with some critical provisions below. For the avoidance of doubts, I am an additional reviewer who had not seen the manuscript previously. My main concern is on the advance and the significance of the paper: it is limited. The manuscript is dealing with a known target for the disease and a known strategy (CAR) for the disease with similar results to what has been reported by other before (Bôle-Richard et al., Leukemia 2020). The strategy of allogeneic TCRko cells has also been reported repeatedly in the past, including clinical trials, so the only novel aspect is the application of this known and well characterized technology, to yet another rare disease. Along these lines the impact and significance to the field is scarce.

Specific comments:

- 1) Analysis of cytotoxicity against CAL1 cells is lacking critical controls, such as TCRKO cells. Also it would have been important to demonstrate absence of toxicity against a panel of unrelated cells.
- 2) It appears that the use of TCRKO cells is not a proper control for the assays: for example sup figure 2, suggests that these cells have in fact a massive defect in effector function as evidenced by limited or lack of cytokine release. A probably more adequate control would have been an irrelevant CAR in this setting.
- 3) The same comment applies to the figure with primary cell lines where the controls for lysis are mentioned in the figure legend but are nowhere to be found.
- 4) The strict dependency on dosage is peculiar, as the different dosages 3 or 10 million are not massive differences, unlike 1 to 10, still the drop in effectivity is sharp. I wonder if this is not a peculiarity of the cell system used here and whether this would occur when benchmarked against an anti-CD123 CAR in primary T cells. This is of particular relevance as the deficiency of the system turns to happen late in the experiments, where likely the short lived edited T cells are already gone.
- 5) Depletion with Rituximab is only partially convincing: in fact the authors show that even depletion concomitant to transfer fails to prevent UCART activity in all mice (see figure of the rebuttal). Similar things are seen in sup figure 10 where rituximab only partially prevents cytokine release, questioning the value of this as a safety switch.
- 6) Also the data on toxicity in vitro (sup figures) and in vivo (rebuttal letter) on healthy hematopoiesis is not convincing. In vitro no toxicity is shown for any condition, so it could also simply mean that the assay was not suited for the task. A control for toxicity (maybe a anti-CD33 CAR) would have been important. Same for the in vivo data, substantial activity is shown against the CD33 compartment for the construct but what does that mean is this substantial or not? This cannot be judged in the absence of controls
- 7) It is at times quite difficult to follow what is actually shown in terms of data? Replicates of what? How are repeats handled, pooling? This must be made cristal clear for each and every figure and subfigure.
- 8) How many donors were used for the generation of the allogeneic products during the course of the study
- 9) Where can sequences of the constructs and targeting sequences be found (patent reference)

REVIEWER COMMENTS

Reviewer #5 (Remarks to the Author):

Cai and colleagues have submitted a manuscript evaluating allogeneic TCR ko anti-CD123 CAR T cells for the treatment of blastic plasmacytoid dendritic cell neoplasm, a rare disease with a mostly poor prognosis. The authors provide in vitro and in vivo evaluation against both cell lines and patient derived models. The manuscript is overall well written and easy to follow. This manuscript has already undergone two rounds of revisions. Most of the data is sound with some critical provisions below. For the avoidance of doubts, I am an additional reviewer who had not seen the manuscript previously. My main concern is on the advance and the significance of the paper: it is limited. The manuscript is dealing with a known target for the disease and a known strategy (CAR) for the disease with similar results to what has been reported by other before (Bôle-Richard et al., Leukemia 2020). The strategy of allogeneic TCRko cells has also been reported repeatedly in the past, including clinical trials, so the only novel aspect is the application of this known and well characterized technology, to yet another rare disease. Along these lines the impact and significance to the field is scarce.

Specific comments:

1) Analysis of cytotoxicity against CAL1 cells is lacking critical controls, such as TCRKO cells.

Response: We apologize for the confusion. All experiments performed included TCR-KO cells group. The data shown in the manuscript represent specific cell lysis which accounts for controls with TCR-KO cells as following.

The viable cell number of UCART123-treated cells was compared to that of samples co-cultured with non-transduced TCR $\alpha\beta$ -deficient (TCR $\alpha\beta$ KO) T cells to determine the percentage of specific cell lysis.

$\%Lysis = (1 - \text{viable cell number of UCART123-treated cells} / \text{viable cell number of TCR}\alpha\beta \text{ KO T cells-treated cells}) \times 100\%$.

To address Reviewer's question, we below show viable cell numbers from each group (Supplementary data Fig. 2a) used for the calculation of specific lysis. TCR-KO cells have negligible effects on viability of CAL1 cells, while UCART123 cells induce dose-dependent cytotoxicity. This data is now included in a new **Supplementary data Fig. 2a**.

Supplementary Fig. 2. Cytotoxicity of UCART123 against CAL-1 BPDCN cells *in vitro*. **a**, CAL-1 BPDCN cells were co-cultured with either non-transduced TCR $\alpha\beta$ -deficient (TCR $\alpha\beta$ KO) T cells or with UCART123 cells at various effector: target (E:T) ratios indicated. After 16 hours co-culture, CAL-1 cell number was quantified and specific cytotoxic activity of UCART123 against CAL-1 target cells was calculated. Each point represents the data obtained from triplicate experiments, and the mean \pm SD value is presented.

Also it would have been important to demonstrate absence of toxicity against a panel of unrelated cells.

Response: In Supplementary data Fig 3, CD123-negative Daudi cell line is included as unrelated control (copied below, **Supplementary data Fig. 3 and Complementary data**). No cytotoxic activity of UCART123 was elicited against Daudi cells, and there was no increased degranulation, or IFN γ release by CART cells.

Supplementary Fig. 3. In vitro activity of T-cells expressing CAR123 against cell lines expressing different levels of CD123. a, The upper row of panel a shows the number of CD123 molecules at the cell surface, using the Qifikit surface antigen quantification kit. Daudi cells were used as a CD123 negative control, while KG1a, MOLM13 and RPMI-8226 were used as targets expressing increasing levels of CD123. The table on the upper right portion shows the mean and the SD values from 6 independent staining. Representative flow cytometry data is shown in the lower row of panel a, with blue histograms corresponding to cells labelled with an anti-CD123 monoclonal antibody (clone 6H6), while the grey histograms correspond to cells labelled with the isotype control. **b,** Panel b shows degranulation activity of CAR123 cells, either cultured alone or co-cultured during 6h with the different cell lines described in panel a, at a 1:1 E:T ratio. Degranulation was evaluated by flow cytometry after staining with CD8 and CD107a antibodies. The degranulation of the CD8+ and CD8- fractions (considered as CD4+ cells) is shown. **c,** Panel c shows IFN γ levels released in the supernatants of overnight co-cultures of CAR123 cells with target cells at 1:1 E:T ratio, using an ELISA test. **d,** Cytotoxic activity against MOLM13 and RPMI-8226 cells is shown in panel d. Co-cultures were settled with each of the CD123+ cell lines (together with Daudi CD123neg cells as an internal negative control) at a 10:1 effector to target (E:T) ratio for 18 hours. Cytotoxic activity was evaluated by assessing the viability of the different target cell populations by flow cytometry at the end of the co-culture. Cell killing activity was normalized to the activity against CD123- Daudi cells. Source data are provided as a Source data file.

NTD: non transduced T-cells. UCART123: T-cells transduced with CAR123, transfected with TRAC targeting TALEN $\text{\textcircled{R}}$ and purified by depletion of remaining TCR $\alpha\beta$ + T-cells. CART123: T-cells transduced with CAR123, mock transfected (TCR $\alpha\beta$ + cells). The experiments shown in panels b-d were performed with T-cells from three different donors, each of which is represented by a different color (black, green or white symbols). The donor highlighted in black was transduced with a R&D backbone, while the two other donors were transduced with an rLV produced from the same backbone used to manufacture UCART123 clinical batches. Nevertheless, all three donors express the CAR from the same EF1 α p-RQR8-2A-CAR123 lentiviral expression cassette.

To further clarify the Reviewer's query, we below include % viable cells for CD123-negative cell line Daudi, and two cell lines with different levels of CD123 (MOLM13 and RPMI-8226), used for the calculation of specific lysis. This demonstrates lack of non-specific cytotoxicity by UCART123

Complementary data for Extended data Fig. 3

Complementary data for Extended Data Figure 3.

In vitro activity of T-cells expressing CAR123 against cell lines expressing different levels of CD123.

Cytotoxic activity against MOLM13 and RPMI-8226 cells is shown in the right panel (Panel D from External Data Figure 3)

Co-cultures were settled with each of the CD123+ cell lines (together with Daudi CD123neg cells as an internal negative control) at a 10:1 effector to target (E:T) ratio for 18 hours.

The data in the histograms on the left show the percentage of viable target cells after the 18h co-culture with TCRab+ Non Transduced T-cells (NTD), TCRab+ CAR123 cells, or UCART123 cells. The viability of each of the target cell lines cultured alone during the same period of time and in the same culture conditions is also shown (target only, gray bars)

cells.

2) It appears that the use of TCRKO cells is not a proper control for the assays: for example sup figure 2, suggests that these cells have in fact a massive defect in effector function as evidenced by limited or lack of cytokine release. A probably more adequate control would have been an irrelevant CAR in this setting.

Response: We appreciate Reviewer's thoughtful comments and concerns.

The cytokine levels (IL-2, IL-5, IL-13 and TNF α) in the group of TCR-KO cells stimulated with PMA/ionomycin were increased compared to untreated cells (Supplementary data Fig. 2b). Most importantly, the IFN γ level (Fig. 1d below) secreted by TCR-KO cells stimulated with PMA/ionomycin is similar with the level in the group of UCART123 cells stimulated with PMA/ionomycin, suggesting acceptable functionality of TCR-KO cells.

Fig. 1. Cytotoxicity of UCART123 against CAL-1 BPDCN cells *in vitro*. d, CAL-1 cells were co-cultured with either TCR $\alpha\beta$ KO T cells or with UCART123 cells at E:T=1:1 for 25 hours. IFN γ levels were determined by the BioLegend LEGENDplex assay. Data represent n=3 biological replicates and mean \pm SD of triplicates. For c, d, Significance was determined using unpaired two-tailed t-test annotated as *** $P < 0.001$. Source data are provided as a Source data file.

3) *The same comment applies to the figure with primary cell lines where the controls for lysis are mentioned in the figure legend but are nowhere to be found.*

Response: As above (see response to question #1), all experiments with primary cells also included TCR-KO cells group. The data shown in the manuscript (Fig. 2) represent specific cell lysis which accounts for controls with TCR-KO cells as explained above. The % of viable cells from each group used for the calculation of specific lysis is now included as revised Supplementary data Fig. 5, and copied for clarity below.

Supplementary Fig. 5. Antitumor activity of UCART123 against primary BPDCN samples *in vitro*. Viability of MOLM13 cells and of the primary BPDCN samples upon co-culture for 16 hours with either UCART123 cells or non-transduced TCR $\alpha\beta$ -deficient (TCR $\alpha\beta$ KO) T cells. Target cells from control group were cultured alone in the same experimental conditions as the co-cultures. Each point represents the data obtained from triplicate experiments, and the mean \pm SD values are shown. Significance was determined using unpaired two-tailed t-test. * $P \leq 0.05$, ** $P \leq 0.01$, *** $P \leq 0.001$. Source data are provided as a Source data file.

4) *The strict dependency on dosage is peculiar, as the different dosages 3 or 10 million are not massive differences, unlike 1 to 10, still the drop in effectivity is sharp. I wonder if this is not a peculiarity of the cell system used here and whether this would occur when benchmarked against an anti-CD123 CAR in primary T cells. This is of particular relevance as the deficiency of the system turns to happen late in the experiments, where likely the short lived edited T cells are already gone.*

Response: We appreciate Reviewer’s thoughtful comments. However, UCART123 cells will not be rejected in immunodeficient mice, and will persist and expand as long as they keep encountering the target antigen. The presence of UCART123 cells was assessed in the blood, spleen, and BM of one mouse from the 10×10^6 UCART123 treatment group sacrificed on day 78 (57 days after UCART123 cell injection). UCART123 cells could be detected in the spleen (16.4% CAR+ cells among viable cells; Supplementary Data Fig. 8, copied below for clarity) and in the bone marrow (1.1% CAR+ cells; Supplementary Data Fig. 8), but we could not detect the UCART123 cells in the low-dose treatment group.

Supplementary Fig. 8. The presence of UCART123 cells was analyzed in the blood, spleen, and bone marrow of the mouse treated with 10×10^6 UCART123 cells (PDX-1 model) and sacrificed on Day 78 (57 days after UCART123 treatment). UCART123 cells were detected by flow cytometry using CD123-Fc protein and an anti-mouse Fc antibody.

5) Depletion with Rituximab is only partially convincing: in fact the authors show that even depletion concomitant to transfer fails to prevent UCART activity in all mice (see figure of the rebuttal).

Response: In the prior Fig 1 (Sugita et al) of the rebuttal copied below, Rituximab was administered not “prior to transfer”, but 7 days after animals received UCART123. This explains incomplete reversal of CART activity, since UCART123 were functional for 7 days prior to depletion. Data shown in **Fig. 1g,h (see below)** demonstrated that treatment with 10mg/kg of rituximab starting at 7 days after animals received UCART123 cells resulted in disease progression evaluated by luminescence, and reduction of detectable CAR T cells by flow cytometry, in contrast with mice that received UCART123 cells and vehicle control (for the rituximab treatment) and remained disease-free. Upon re-challenging mice with leukemia cells, mice treated with rituximab had significantly lower overall survival and more rapid disease progression compared with UCART123-pretreated mice that received vehicle and no Rituximab (**Supplementary Data fig.1e-f** in co-submitted manuscript, not shown here), demonstrating that rituximab eliminated residual UCART123.

Fig. 1 (Sugita et al)

Fig. 1 (Sugita et al). g, MOLM13-BLIV engrafted mice were treated with PBS, UCART123 cells, or UCART123 cells followed by rituximab (RTX). Representative BLI images at pre-treatment, day 28 and day 42 measured are shown for each group. h, top, Frequencies of UCART123 cells (%) in mononuclear cells in peripheral blood on day 23 post UCART123 treatment were measured with flow cytometry. Each symbol represents one mouse and bar represents the average with the SD. *** $p < 0.0001$, one-way ANOVA. bottom, Average radiance measured with BLI on day 28 and day 42 are shown. Each symbol represents one mouse and bar represents the average with the SD. *** $p < 0.001$, Mann-Whitney test.

Similar things are seen in sup figure 10 where rituximab only partially prevents cytokine release, questioning the value of this as a safety switch.

Response: We concur with the Reviewer that rituximab partially reduces cytokine release in this study (**Supplementary data Fig. 10**). It should be noted that the cytokine level was similar on Day 19 prior to Rituximab therapy; and drastically decreased on Day 23 after the Rituximab ($1,802 \pm 517$ pg/mL vs $8,976 \pm 2,539$ pg/mL), which should translate in abrogating severe CRS if observed in the clinical setting. In this study, Rituximab at 10mg/kg was administered i.p. 2 days after UCART injections (day 19) for a total of 5 days. It is possible that the high dose and/or longer therapy will completely abrogate UCART activity.

Supplementary Fig. 10. Rituximab depletes UCART123 *in vivo*. **a**, Experimental design using BPDCN-3 PDX cells. When engraftment was confirmed on day 17 after tumor cell injection, mice were randomized into 6 treatment groups ($n=9$ mice/group) and received treatment as follows: vehicle; 10×10^6 TCR $\alpha\beta$ KO T cells; 3×10^6 UCART123 cells; 3×10^6 UCART123 cells followed by Rituximab; 10×10^6 UCART123 cells or 10×10^6 UCART123 cells followed by Rituximab. Rituximab at 10mg/kg was administered i.p. 2 days after UCART injections (day 19) for a total of 5 days. **b**, Fractions of UCART123 cells in the spleen and bone marrow of mouse ($n=3$) from experimental cohorts. Mice were sacrificed on day 24 after tumor cell injection. UCART123 cells were detected by flow cytometry using CD123-Fc protein and an anti-mouse Fc antibody conjugated with PE. The mean \pm SD value is presented. **c**, IFN- γ levels in peripheral blood of mice measured on day 19 and day 23. The mean \pm SD value is presented. Significance was determined using unpaired two-tailed t-test. Compared to 10×10^6 UCART123 cells group: *** $P < 0.001$. Source data are provided as a Source data file.

6) Also the data on toxicity *in vitro* (sup figures) and *in vivo* (rebuttal letter) on healthy hematopoiesis is not convincing. *In vitro* no toxicity is shown for any condition, so it could also simply mean that the assay was not suited for the task. A control for toxicity (maybe a anti-CD33 CAR) would have been important. Same for the *in vivo* data, substantial activity is shown against the CD33 compartment for the construct but what does that mean is this substantial or not? This cannot be judged in the absence of controls

Response: We concur with the Reviewer that inclusion of an additional control, for example anti-CD33 CAR, would be helpful to demonstrate toxicity in the *in vitro* and *in vivo* assays on healthy hematopoiesis. This could be considered in future studies. However, it is important to point out that (1) every CAR construct will have different cytotoxicity and activity profile, (2) we utilized the same assays and the same doses of UCART123 and control T cells for assessment of the *in vitro* cytotoxicity and showed reduction of BPDCN. We observed a dose dependent toxicity as expected in myeloid colony formation of healthy donor cells (when using much higher E:T ratios) but no impact on normal progenitor growth when using lower E:T ratios that are effective against BPDCN cells (Supplementary data Fig. 6).

Supplementary Fig. 6. The toxicity of UCART123 against normal hematopoietic cells *in vitro*. **a**, Normal BM-derived hematopoietic cells were co-cultured with either UCART123 cells or non-transduced TCR $\alpha\beta$ -deficient T cells (TCR $\alpha\beta$ KO) for 16 hours and counted by flow cytometry. Each point represents the data obtained from triplicate experiments, and the mean \pm SD value is shown. **b**, Normal BM-derived hematopoietic stem cells were co-cultured for 2 weeks with UCART123 at different ratios and the cultures analyzed by colony-formation assay. Erythroid colony-forming units (CFUs) and myeloid CFUs were counted separately. Each point represents the data obtained from triplicate experiments, and the mean \pm SD value is presented. Source data are provided as a Source data file.

7) It is at times quite difficult to follow what is actually shown in terms of data? Replicates of what? How are repeats handled, pooling? This must be made crystal clear for each and every figure and subfigure.

Response: We concur with the Reviewer that we should clarify how are repeats handled. For the *in vitro* assays using cell line, we independently repeated the experiment at least twice. For the *in vitro* assays using primary cells, due to rarity of the disease and limited cell numbers, we performed the experiment once using triplicate wells. We collected 8 primary samples for the assays to increase the rigor of the data and used independent samples as biological replicates. For the *in vivo* study, we performed experiments using three separate BPDCN PDX models (Fig. 3, 4 and 5).

8) How many donors were used for the generation of the allogeneic products during the course of the study

Response: UCART123 cells used in this study were derived from a single donor. Two large scale batches (derived from the same donor) were used for *in vitro* testing, while a single batch was used in all the *in vivo* experiments.

For the data presented in Supplementary Fig 3, three small scale batches were used, from three independent donors.

We have further added in the Methods the following:

UCART123 production

R&D grade UCART123 cells were produced by Collectis using a large-scale manufacturing process. UCART123 cells were derived from a single donor. Two large scale batches (derived from the same donor) were used for *in vitro* testing, while a single batch was used in all the *in vivo* experiments. For the data presented in Supplementary Fig 3, three small scale batches were used, from three independent donors.

9) Where can sequences of the constructs and targeting sequences be found (patent reference)

Response: The CAR construct used is described in patent EP3119807B1 (<https://patents.google.com/patent/EP3119807A1/en>).

We used the scFv derived from the monoclonal antibody Klon43 fused to CD8a hinge and transmembrane domains, the 4-1BB co-stimulatory domain, and the CD3zeta signaling domain.

We have further added in Methods the following:

CAR construct

The CAR construct used is described in patent EP3119807B1.

We used the scFv derived from the monoclonal antibody Klon43 fused to CD8a hinge and transmembrane domains, the 4-1BB co-stimulatory domain, and the CD3zeta signaling domain.